# Increasing Information Extraction in Low-Signal Regimes via Multiple Instance Learning

## Abstract

In this work, we introduce a new information-theoretic perspective on Multiple Instance Learning (MIL) for parameter estimation with i.i.d. data, and show that MIL can outperform single-instance learners in low-signal regimes. Prior work (Nachman & Thaler, 2021) argued that single-instance methods are often sufficient, but this conclusion presumes enough single-instance signal to train near-optimal classifiers. We demonstrate that even state-of-the-art single-instance models can fail to reach optimal classifier performance in challenging low-signal regimes, whereas MIL can mitigate this sub-optimality. As a concrete application, we constrain Wilson coefficients of the Standard Model Effective Field Theory (SMEFT) using kinematic information from subatomic particle collision events at the Large Hadron Collider (LHC). In experiments, we observe that under specific modeling and weak signal conditions, pooling instances can increase the effective Fisher information compared to single-instance approaches.

## 1 Introduction

Hypothesis testing provides a formal framework for deciding between a null hypothesis $H_0$ and an alternative hypothesis $H_1$, based on observed data. According to the Neyman-Pearson lemma (Neyman et al., 1997), the uniformly most powerful test statistic is the log-likelihood ratio (LLR), making it the optimal choice for distinguishing between competing models. Equation 1 shows the LLR when the number of events observed, $N$, is a Poisson random variable with mean rate $\nu(\theta)$. As shown, the LLR, denoted as $\Lambda(x|\theta_1, \theta_0)$, depends on the data $x$, the parameter of interest $\theta$, the mean rate $\nu$, and event likelihood $p(x_i|\theta)$.

$$\Lambda(x|\theta_1,\theta_0) = \underbrace{(\nu(\theta_0) - \nu(\theta_1) + N\ln(\frac{\nu(\theta_1)}{\nu(\theta_0)}))}_{\text{Rate Term}} + \underbrace{\sum_{i=1}^{N}\ln(\frac{p(x_i|\theta_1)}{p(x_i|\theta_0)})}_{\text{Shape Term (ML Target)}} \tag{1}$$

In high-energy particle physics, we usually have a well-defined means of calculating the expected event rate under some hypothesis, but the likelihood $p(x_i|\theta)$ for a single event is often intractable. (Brehmer et al., 2020) One common strategy is to generate Monte-Carlo simulations (Frederix et al., 2021) under different parameter values, and train ML models to approximate a function monotonically related to the LLR, using kinematic observables such as energy and momentum as input. However, the practical effectiveness of this method diminishes when the underlying signal is weak, causing even state-of-the-art classifiers to exhibit suboptimal performance in practice.

Since the signal levels are lower than for a given ML model to construct a reliable discriminant, we conceived the use of set (or "bag") of events in order to aggregate the faint signals to strong, coherent signatures. This set-based approach is conceptually related to what is colloquially known as "Multiple Instance Learning" (MIL), but it differs fundamentally in its objective.

The MIL is a form of data fusion (Lip & Ramli, 2012) and weakly supervised learning, where instead of each instance having its own label there is only a single label for a set (or a "bag") of instances. For example, as it was first proposed by Dietterich et al. (1997) for drug activity prediction, in a

binary classification problem, the bag would be labeled as positive if there is at least one positive case in the bag, and it would be labeled as negative if all instances are negative. The MIL has many use cases such as medical image analysis (Quellec et al., 2017), object detection (Yuan et al., 2021), image classification (Rymarczyk et al., 2021), and many others; a comprehensive review can be found in the work of Waqas et al. (2024).

In our work, the objective of the MIL classifier is not to identify a single "key" instance, but, as we will show in Section 3, to aggregate the subtle statistical signature that is distributed across every instance in the bag. While prior work has developed related approaches in both the weakly supervised and multi-event settings, our emphasis and results differ. In particular, the Classification without Labels (CWoLa) paradigm (Metodiev et al., 2017) establishes that classifiers trained on mixed samples can recover the optimal fully supervised classifier under idealized conditions (i.e. sufficient amount of data and signal fraction). Likewise, Nachman & Thaler (2021) analyzed connections between per-instance and per-ensemble classifiers and demonstrated constructive mappings between them under IID assumptions.

However, in low-signal regimes, the equivalence implied by those theoretical constructions can fail in practice as the classifiers become suboptimal. Therefore, in this paper, we bring an information-theoretic perspective to the previous multi-event literature, and identify conditions under which set-based aggregation improves inference. To our knowledge, no prior work has rigorously characterized MIL's impact on hypothesis testing in low-signal regimes, especially in the context of Fisher Information estimation. Concretely, our main contributions are:

- We provide a mathematical motivation for why MIL can help mitigate sub-optimality in low-signal regimes, and we derive how aggregation affects the *effective* Fisher information, thereby pushing the precision of parameter measurements closer to its theoretical limit.

- We present a counterexample equivalence between single-instance and multi-instance learners, and demonstrate that under certain low-signal, finite model/data regimes, MIL can yield better performance than single-instance learners that were previously assumed to be sufficient.

- We identify that learned models violate the second Bartlett identity (Bartlett, 1953), therefore we provide a practical post-hoc calibration procedure to address this issue.

- We investigate the performance of this framework across multiple settings, providing insights into their respective strengths and limitations for this high-precision task.

The remainder of this paper is organized as follows: Section 2 provides the necessary background on our analysis case, giving a brief introduction to Standard Model (SM), and Standard Model Effective Field Theory (SMEFT) parameters. Section 3 details our theoretical framework, and in Section 4 we provide our preliminary results to show under which conditions our results can be aligned with our theoretical predictions. Finally, in Section 5 we briefly summarize our research, and share our ideas for future work.

## 2 HYPOTHESIS STUDY: STANDARD MODEL VS STANDARD MODEL EFFECTIVE FIELD THEORY

SMEFT provides a consistent quantum field theory framework that parameterizes the low-energy effects of new, high-energy phenomena on the known SM fields. This is achieved by extending the SM Lagrangian ($\mathcal{L}_{\text{SM}}$) with a series of higher-dimensional operators, $\mathcal{O}_i$:

$$\mathcal{L}_{\text{SMEFT}} = \mathcal{L}_{\text{SM}} + \sum_i \frac{c_i}{\Lambda^{d_i-4}} \mathcal{O}_i, \qquad (2)$$

Where $\Lambda$ is the new physics scale at which degrees of freedom are integrated out, leaving their low-energy effects encoded in effective operators. In essence, the Wilson coefficients quantify the strength of new, unobserved interactions; a nonzero value for any $c_i$ would indicate a deviation from the SM and thus be a sign of new physics.

While there are many Wilson coefficients affecting different particle interactions, the goal of this paper is not to perform a comprehensive physics analysis, but to analyze the behavior of the analysis

tools themselves. Therefore, in this paper we will only focus on a single type of particle interaction as what physicists call "signal" events, i.e. the collision events which are sensitive to the new physics parameter. For our analysis, we choose to focus on Higgs to WW boson decay channel as our signal events, with the Wilson coefficient value $c_{HW}$ is set to a non-zero value. For our "background" events, we used Higgs to ZZ boson decay channel with no SMEFT effects. These background events are not influenced by the parameters of interest, but have similar experimental signatures to the signal, acting as a form of noise that complicates the classification task. Further details on the simulation process are provided in the Appendix B.1.

The analysis thus simplifies to a hypothesis test problem: a value of $c_{HW} = 0$ corresponds to the SM, while $c_{HW} \neq 0$ indicates physics beyond the SM. As it is detailed in Appendix B.2, we kept our implementation as simple as possible in order to make our analysis a general hypothesis testing problem. We analyzed the behavior of ML models in three different settings:

1. **Binary Classification:** Distinguishing between SM ($c_{HW} = 0$) and SMEFT ($c_{HW} \neq 0$) hypothesis using event kinematics.

2. **Multi-Class Classification:** Using event kinematics to predict the specific value of $c_{HW}$ from a discrete set of possibilities.

3. **Parameterized Neural Networks** (Baldi et al., 2016): Training a neural network that takes both the event kinematics $x$ and parameter value $\theta$ as input, i.e. $[x, \theta]$, and determine if the kinematics are consistent with that specific parameter value. After training, one can continuously change the $\theta$ value to find the "best match" for a given kinematic input.

## 3 MOTIVATION FOR MULTIPLE INSTANCE LEARNING FOR HYPOTHESIS TESTS

In this section we mathematically derive **(i)** how MIL increases information content per prediction, and **(ii)** how a decrease in ML error, or an increase in model optimality, affects the Fisher Information extracted from the data. To simplify our mathematical arguments, we will focus on a single physical parameter of interest, though the argument can be readily generalized to an arbitrary number of parameters.

### 3.1 DISTINGUISHING THE INDISTINGUISHABLE

Let $\mathbf{x} \in \mathcal{X} \subseteq \mathbb{R}^d$ be a vector containing single instance of high-energy particle collision event information, and $\theta \in \Theta \subset \mathbb{R}^p$ be the parameter of interest. The probability density function (PDF) of observing event $\mathbf{x}$ given parameters $\theta$ is denoted by $p(\mathbf{x}|\theta)$.

The collision events are independent and identically distributed (i.i.d.), therefore the joint probability of a set of events $\{\mathbf{x_i}\}_{i=1}^N$ under a model parameterized by $\theta$ (SM or SMEFT) is the product of individual event probabilities is,

$$p(\{\mathbf{x_i}\}_{i=1}^N|\theta) = \prod_{i=1}^N p(\mathbf{x_i}|\theta). \tag{3}$$

The SM would correspond to $\theta_{SM} = 0$, while the SMEFT would correspond to $\theta_{SMEFT} \neq 0$. For small deviations from SM we can define a perturbation $\delta p(\mathbf{x_i})$, where $\delta p(\mathbf{x_i}) \ll p(\mathbf{x_i}|\theta_{SM})$, such that the likelihood ratio of a given event $r(\mathbf{x_i})$ would be,

$$r(\mathbf{x_i}) \approx \frac{p(\mathbf{x_i}|\theta_{SM}) + \delta p(\mathbf{x_i})}{p(\mathbf{x_i}|\theta_{SM})} = 1 + \frac{\delta p(\mathbf{x_i})}{p(\mathbf{x_i}|\theta_{SM})}. \tag{4}$$

Taylor expanding the log-likelihood ratio, denoted by $\lambda_i(\mathbf{x_i}|\theta_1, \theta_0)$, would give some small $\eta_i$:

$$\lambda_i(\mathbf{x_i}|\theta_1, \theta_0) \approx \frac{\delta p(\mathbf{x_i})}{p(\mathbf{x_i}|\theta_{SM})} = \eta_i \tag{5}$$

Now, this might be problematic for an Event-By-Event (*EBE*) classifier, because in order to make an accurate prediction the ML model has to accurately discern the small $\eta_i$ values for different

samples, each treated as an independent case. On the other hand, for a bag of events $\mathcal{B} = \{\mathbf{x_i}\}_{i=1}^{N}$ the information available to bag-level (*BAG*) classifiers is:

$$\frac{1}{N} \ln r(\mathcal{B}) = \frac{1}{N} \sum_{i=1}^{N} \eta_i \to \mu_\eta, \qquad\qquad \ln r(\mathcal{B}) \approx N\mu_\eta \qquad (6)$$

To understand why bag-level classifiers are able to discern the observed data that the event-level classifiers fails to distinguish from each other, we can take a look at the Signal-to-Noise Ratio (SNR = $\mu/\sigma$) of the inputs. Assuming homogeneity, since events are independent $\mathrm{Var}(\ln r(\mathcal{B})) = N\,\mathrm{Var}(\eta_i) = N\sigma_\eta$,, and the SNRs are:

$$\mathrm{SNR_{BAG}} = \frac{|\mathbb{E}[\ln r(\mathcal{B})]|}{\sqrt{\mathrm{Var}(\ln r(\mathcal{B}))}} = \frac{N|\mu_\eta|}{\sqrt{N\sigma_\eta^2}} = \boxed{\sqrt{N}\,\frac{|\mu_\eta|}{\sigma_\eta}} \qquad (7)$$

$$\mathrm{SNR_{EBE}} = \frac{|\mathbb{E}[\ln r(x)]|}{\sqrt{\mathrm{Var}(\ln r(x))}} = \boxed{\frac{|\mu_\eta|}{\sigma_\eta}} \qquad (8)$$

We see that the SNR increases with $\sqrt{N}$ for the bag-level classifiers, meaning that MIL provides increasing discriminative information as $N$ grows, even if the individual $\eta_i$ are small. As demonstrated by Nachman & Thaler (2021), bag-level and event-level predictors should produce the same results in the idealized i.i.d. setting because of the mathematical equivalence in Eq. 3. However, as we discuss in Section 4.1 and Appendix C.3, when the SNR is below a certain threshold learned models can fail to reach optimal discriminator performance given finite data. Since MIL increases the SNR, it can mitigate these finite sample/model-induced sub-optimality; therefore, MIL can improve performance and cause a practical breakdown of the theoretical equivalence. Section 4 presents empirical results that align with these predictions.

## 3.2 Increasing the effective Fisher Information

In essence, bag-level classifiers create summary statistics. The specific implementation of this summarization is left to the machine learning practitioner. Rather than showcasing the capabilities of some unique architecture, we utilized a basic neural network model to demonstrate the power of this methodology. The basic implementation is as follows:

- Use 3 layer, 64 neuron Multi-Layer Perceptron as an embedding function $\phi(\mathbf{x_i})$ which takes the feature vector $\mathbf{x_i}$ and maps it to an embedding vector $\mathbf{e_i}$.
- Take the average of the embedding vectors in a given bag: $\bar{\mathbf{e}} = \frac{1}{N} \sum_i^N \mathbf{e_i}$
- The logit of the final layer in the binary classifier and the log-probability ratio of multi-class classifier would yield the log-likelihood ratio $\Lambda$ of the whole bag. (see Appendix B)

We would like to emphasize, we are *not* taking the average of the probabilities. We are taking the average of the embedding vectors, in order to create what we call *Asimov Vector* $\bar{\mathbf{e}}$. [1] The goal is to create an amalgamation of all of the events contained in the bag for a single prediction.

Now, let $\lambda_{\text{true}}(\mathbf{x_i}|\theta_1, \theta_0)$ be the true value of the LLR of a single event $\mathbf{x_i}$, and $\Lambda_{\text{true}}(\mathcal{B}_j|\theta_1, \theta_0)$ be the true LLR value for the bag of events $\mathcal{B}_j$, with number of events in the bag denoted by $N_B$. The true LLR $\Lambda_{\text{true}}(\mathcal{B}_j)$ would be the sum of the event LLRs $\lambda_{\text{true}}(\mathbf{x_{jk}})$,

$$\Lambda_{\text{true}}(\mathcal{B}_j) = \sum_{k=1}^{N_B} \lambda_{\text{true}}(\mathbf{x_{jk}}) \qquad (9)$$

The ML model's prediction $\hat{\Lambda}_j = \Lambda_{\text{true}}(\mathcal{B}_j) + \epsilon_j$, would have an error $\epsilon_j$. Samples are independent collision events, and for unbiased estimate of the $\Lambda_{\text{true}}(\mathcal{B}_j)$, the expected value is $\mathbb{E}_\theta[\epsilon_j] = 0$. But

---

[1]Asimov Vector is named after Asimov Dataset (Cowan et al., 2011) which is named after Isaac Asimov, the author of the short story *Franchise*. In the story, the super-computer Multivac selects a single representative voter for the entire population, avoiding the need for an actual election.

the variance $\text{Var}_{\theta_0}(\epsilon_j) = \sigma_\epsilon^2(N_B)$ may be a function of $N_B$, the bag size. The test statistic $T$ for the entire dataset $D$, with M number of bag of events would be:

$$T(D) = \sum_{j=1}^{M} \hat{\Lambda}(\mathcal{B}_j) = \sum_{j=1}^{M}(\Lambda_{\text{true}}(\mathcal{B}_j) + \epsilon_j) = \Lambda_{\text{true, dataset}}(D) + \sum_{j=1}^{M} \epsilon_j \qquad (10)$$

If we define $I_B(\theta_0)$ as the Fisher Information of a bag of events, through similar calculations stated in Appendix A, one can show that

$$\mathbb{E}_{\theta_1}[T] \approx +\frac{1}{2}MI_B(\theta_0)(\Delta\theta)^2 \qquad\qquad \mathbb{E}_{\theta_0}[T] \approx -\frac{1}{2}MI_B(\theta_0)(\Delta\theta)^2 \qquad (11)$$

And the total variance $T(D)$ under $\theta_0$ would be,

$$\text{Var}_{\theta_0}(T(D)) \approx MI_B(\theta_0)(\Delta\theta)^2 + M\sigma_\epsilon^2(N_B) \qquad (12)$$

To study the relationship between the information latent in the dataset and the information extractable by ML models, we calculate the $\text{SNR}^2 = (\Delta\mathbb{E}[T(D)])^2/\text{Var}_{\theta_0}(T(D))$. This relates the true Fisher Information of the whole dataset, $I_{\text{true, D}}(\theta) = MI_B(\theta_0)$, to the *effective* Fisher Information of the whole dataset, $I_{\text{eff, D}}(\theta)$:

$$I_{\text{eff, D}}(\theta_0)(\Delta\theta)^2 \approx \frac{(MI_B(\theta_0)(\Delta\theta)^2)^2}{MI_B(\theta_0)(\Delta\theta)^2 + M\sigma_\epsilon^2(N_B)} \qquad (13)$$

$$I_{\text{eff, D}}(\theta_0) \approx \frac{M^2 I_B(\theta_0)^2(\Delta\theta)^2}{MI_B(\theta_0)(\Delta\theta)^2 + M\sigma_\epsilon^2(N_B)} = \frac{MI_B(\theta_0)}{1 + \frac{M\sigma_\epsilon^2(N_B)}{MI_B(\theta_0)(\Delta\theta)^2}} \qquad (14)$$

$$\boxed{I_{\text{eff, D}}(\theta_0) = \frac{I_{true,D}(\theta)}{1 + \frac{\sigma_\epsilon^2(N_B)}{I_B(\theta_0)(\Delta\theta)^2}} = \frac{I_{true,D}(\theta)}{1 + \frac{\sigma_\epsilon^2(N_B)}{N_B I_1(\theta_0)(\Delta\theta)^2}}} \qquad (15)$$

Here, $I_1(\theta_0)$ denotes the Fisher Information of a single event, such that $I_B(\theta_0) = N_B I_1(\theta_0)$. Since the calculations are similar in nature to the previous part, we would like to make a distinction: In Equation 15, $\sigma_\epsilon^2(N_B)$ refers to the variance of the ML model's estimation error, not the variance of the log-likelihood ratios themselves.

The equation for the effective Fisher Information we derived captures the asymptotic behaviors of neural estimators in different signal regimes. In the high-signal regime, the error term in the denominator is negligible, and $I_{\text{eff}}(\theta) \approx I_{true}(\theta)$. Conversely, in the low-signal regime, the information contained in a single sample, $I_1(\theta_0)$, is low while the variance of the ML-induced error, $\sigma_\epsilon^2(N_B)$, is high; thus, their ratio becomes non-negligible when $N_B = 1$, reducing the effective information.

Furthermore, if the ML models are well behaved and consistent in results, one can profile the $\sigma_\epsilon^2(N_B)$ function with respect to $N_B$, and extrapolate the amount of increase or decrease of effective Fisher Information. For desirable cases when $\sigma_\epsilon^2(N_B)$ is a sublinear function of $N_B$, the effective Fisher Information would increase as one scales $N_B$. By profiling this behavior, it may very well be possible to extrapolate the *True* Fisher Information. Moreover, for an unbiased estimator of $\theta$, the $\hat{\theta}(D)$, the Cramér–Rao bound (Rao, 1992) on the variance is,

$$\boxed{\text{Var}_\theta(\hat{\theta}) \geq \frac{1}{I_{\text{true, D}}(\theta)}} \qquad (16)$$

Therefore, one of the primary objectives of phenomenological studies, finding the tightest bounds on a given parameter of interest may be achieved through this methodology. Since the standard error of an efficient (or asymptotically efficient) estimator $\hat{\theta}$ is approximately $\sqrt{1/I_{\text{eff, D}}(\theta)}$, as $I_{\text{eff, D}}(\theta)$ approaches $I_{\text{true, D}}(\theta)$, the standard error of our effective estimator approaches its theoretical minimum. Since the width of confidence intervals is proportional to the standard error, maximizing the effective Fisher Information leads to the statistically tightest possible confidence intervals for the parameters of interest.

# 4 RESULTS

## 4.1 BINARY CLASSIFICATION

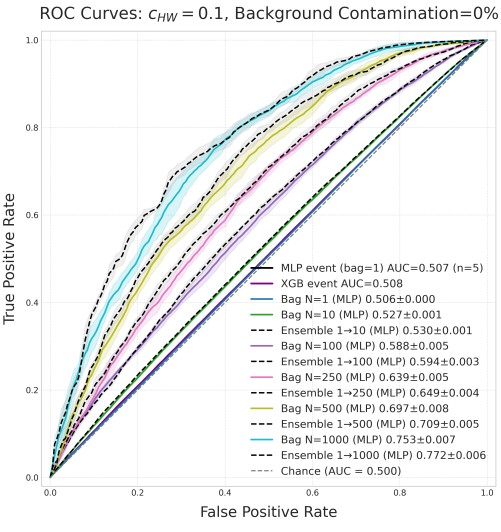
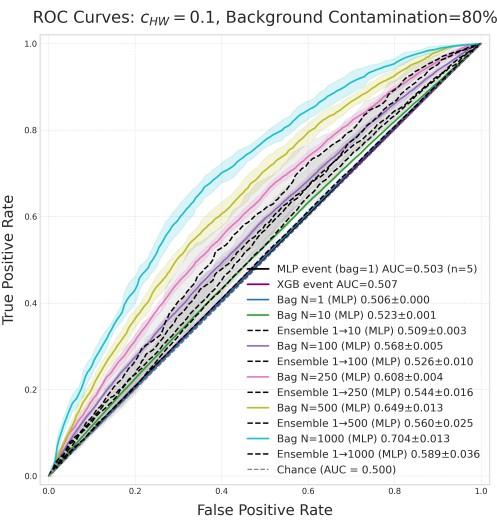

(a) Signal events + 0% Background events

(b) Signal events + 80% Background events

Figure 1: Receiver Operating Characteristic (ROC) curves for binary classification of SMEFT ($c_{HW} = 0.1$) vs. SM with different levels of background event contamination with respect to number of signal events in the bag. Additional contamination levels are shown in Figure 6.

To empirically validate the practical breakdown of the theoretical equivalence between event-level and bag-level predictions, we designed a binary classification task in a challenging, low-signal regime. The ML models are tasked with differentiating between SM vs. SMEFT "signal events" (i.e., events influenced by the parameter of interest) while background events are injected as additional noise. For intuitive visualization of the results, we held the number of signal events in each bag constant and increased the total bag size as we scaled the background contamination level. For example, a bag with 100 signal events and 20% background contamination contains 120 events in total, while 40% contamination corresponds to 140 events. Since background events do not provide any useful information, an optimal discriminator should yield the same ROC curves across all background contamination levels.

For each bag size we trained five Multi-Layer Perceptron (MLP) models with different initializations seeds values. We also constructed ensemble predictions from the event-level classifiers (see Appendix C.3 for details of this procedure) and compared those to the bag-level classifiers. The details of training and optimization can be found in Appendix B.2. Figures 1 and 6 show that the event-level models do not behave optimally in the low-SNR regime: ROC-AUC systematically decreases as background contamination increases, while the bag-level (MIL) classifiers retain substantially better discriminative performance. Furthermore, we also trained a hyperparameter-optimized XGBoost model (Chen & Guestrin, 2016), a strong baseline for tabular data (Shwartz-Ziv & Armon, 2022), and observed a similar scaling behavior with respect to the SNR (Figure 7). Although XGBoost outperforms the simple MLP at relatively high SNR, MIL performance can match or even exceed that of XGBoost in the low-SNR regime. This model-independent degradation of event-level performance, together with MIL's resilience, validates our arguments stated in Section 3.1.

## 4.2 MULTI-CLASS CLASSIFICATION

After investigating completely independent LLR prediction values at discrete $c_{HW} = \theta_k$ values using binary classifiers (see Appendix C.3), we move to multi-class classification in order to couple the LLR predictions and investigate the model's ability to perform precise parameter estimation.

This task imposes stricter requirements on the learned likelihood approximation. For a parameter estimator to follow the frequentist view of confidence intervals, two requirements must be met:

1. The maximum likelihood estimate point must vary with the inherent statistical variance of the data. Let the Fisher Information calculated from the variance of maximum likelihood estimate $\hat{\theta}$ be $I_{\text{MLE}} = 1/\operatorname{Var}(\hat{\theta})$.

2. Since LLR, the $\Lambda$, is asymptotically $\chi^2$ distributed, concavity of the $\Lambda(D, \theta) \approx \Lambda(D, \hat{\theta}) - \frac{1}{2}(\theta - \hat{\theta})^2 I_{\text{curv}}$ must be also equal to the Fisher Information.

As a consequence of the second Bartlett identity (Bartlett, 1953), we know that an ideal, efficient estimator must satisfy these two conditions, since they are the measurement of the same Fisher Information: $I_{\text{true}} \approx I_{\text{MLE}} \approx I_{\text{curv}}$.

But our empirical investigation revealed an unexpected finding: the learned LLR function from our simple MLP model systematically violates the second Bartlett identity, even for event-by-event classifiers. While the location of the LLR minimum correctly tracks the maximum likelihood estimate ($\hat{\theta}$), the learned curvature is consistently too shallow. This result means that the neural network produces an estimator where the information contained in the variance of its score is greater than the information contained in its average curvature, in other words $I_{\text{curv}} < I_{\text{MLE}}$, and

$$\mathbb{E}\left[-\frac{d^2 T}{d\theta^2}\right] < \operatorname{Var}\left(\frac{dT}{d\theta}\right) \tag{17}$$

This underestimated curvature leads to confidence intervals that are too broad, resulting in significant over-coverage (e.g., coverage exceeding 90% at the $1\sigma$ level). To address this, we introduce a single empirically determined calibration constant, $c_{\text{cicc}}$ which rescales the LLR curvature to restore correct frequentist coverage. After this one-time calibration, the $1\sigma$ confidence intervals correctly covered the true parameter value in $68.3 \pm 0.2\%$ of pseudo-experiments. The details of this procedure and the resulting values are provided in the Appendix C.

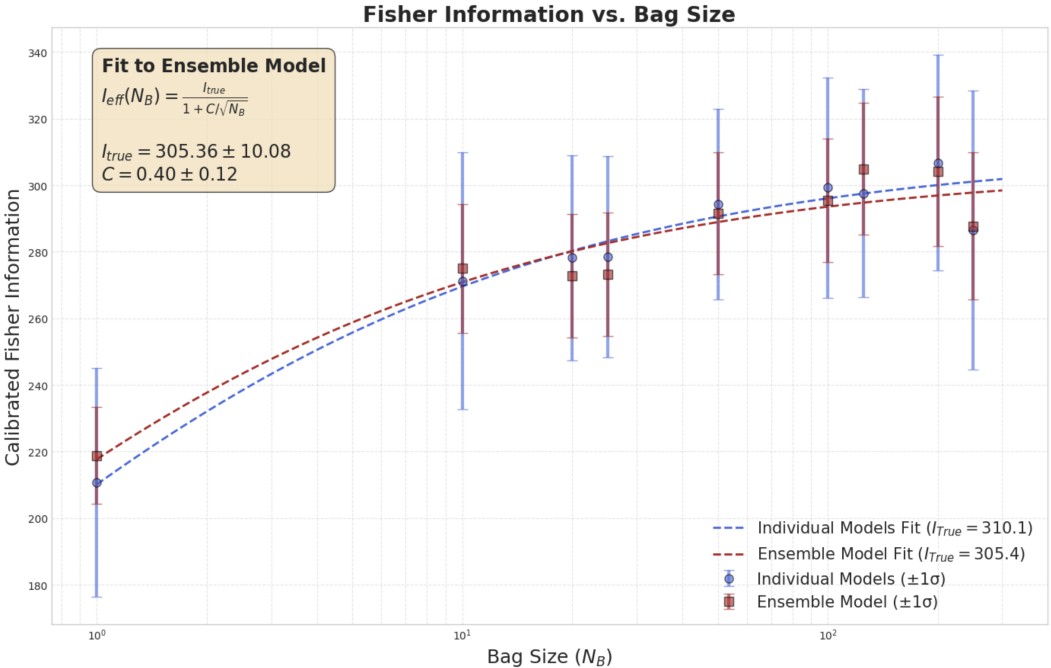

Figure 2: The increase in effective Fisher Information with respect to bag size. Since different 1000 event chunks contain different levels of Fisher Information, the $1\sigma$ variation of information contained in different bags is also showcased with the bars.

To demonstrate the performance of this approach, we constructed 200 confidence intervals from 1000-event data chunks, using 20 ML models trained under identical settings with different initialization seed values. As it is detailed in Appendix C.4, to make the predictions of bag-level classifiers with bag level $< 10$ more robust, by taking average of the predictions of 20 different ML models, we also created what is called an ensemble model. The increase in effective Fisher Information with respect to the bag size is shown in Figure 2. For *illustrative purposes*, we fit the data using a simple model for the error variance of the form $\sigma_\epsilon^2(N_B) = C\sqrt{N_B}$, where $C$ is a free parameter of the fit. Even with this simplified ansatz solution of $\sigma_\epsilon^2(N_B)$, the apparent increase in effective Fisher Information, and its diminishing return with respect to bag size shows another clear and strong evidence supporting our theoretical claims.

### 4.3 PARAMETERIZED NEURAL NETWORKS

Finally, we investigated an alternative architecture, the Parameterized Neural Network (PNN), for the parameter estimation task. Despite extensive experimentation on hundreds of training runs with various stabilization techniques (see Appendix C.5), we found that PNNs, *in their standard implementation*, are not well suited for this high-precision inference task.

Various aspects of PNN contribute to additional deviations from the true value in addition to the models' usual error. For example, the unconstrained nature of the PNN output often led to LLR shapes devoid of any physical meaning, such as smoothed step functions rather than the expected parabolic form. Furthermore, by its design, the output probabilities over the parameter of interest, $c_{HW}$, are not normalized. Therefore, because of the nonlinearity of logit function in the LLR calculations, the outliers create a disproportionate effect on the final decision where the maximum likelihood estimate is, and the curvature of the LLR. Straightforward attempts to mitigate these issues, for example, by artificially normalizing probabilities over the $c_{HW}$ values, did not lead to stable or improved performance.

As shown in Figure 3a, the resulting ML error term, $\sigma_\epsilon^2(N_B)$, did not show a consistent or well-behaved scaling with the bag size. We conclude that while the standard PNNs are effective for other inference tasks, their architectural design may lack the necessary constraints and robustness for the high-precision, curvature-sensitive measurements central to this work.

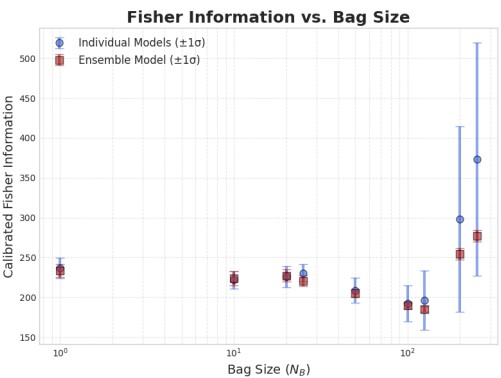
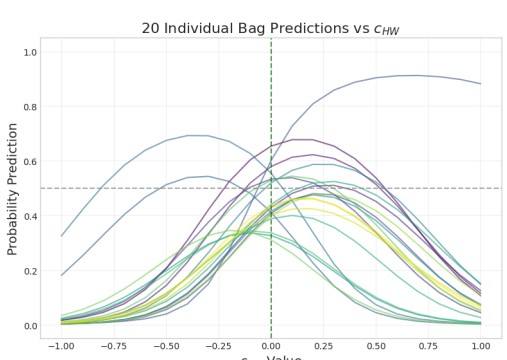

(a) Scaling behavior of effective Fisher Information with respect to bag size.

(b) 20 individual bag probability predictions, $p(B_i|\theta)$, with respect to $\theta$.

Figure 3: Inconsistent and unphysical predictions of Parameterized Neural Networks.

### 5 CONCLUSIONS AND OUTLOOK

In this work we presented a new information-theoretic perspective on Multiple Instance Learning (MIL) for parameter estimation with i.i.d. data and validated our predictions by demonstrating a practical breakdown of the theoretical equivalence between single-instance and multiple-instance learners in low-signal regimes. Our analysis complements prior weakly-supervised and multi-event

results (e.g., Metodiev et al. (2017) and Nachman & Thaler (2021)) by identifying concrete finite-model and finite-sample mechanisms that can make aggregation beneficial in practice.

Our main contributions and findings are:

1. We developed an analytical framework that motivates the set-level aggregation strategy, showed that the effective signal-to-noise ratio can scale like $\sqrt{N_B}$, and derived an expression that relates the model's performance to the Fisher Information available in the dataset under explicit assumptions.

2. We provided empirical evidence supporting the theory:
   - We demonstrate that the SNR increase from aggregation makes MIL more resilient to performance degradation than its single-instance counterparts in low-signal regimes, providing a concrete counterexample to the asymptotic equivalence between single-instance and multiple-instance learners under finite-data/model conditions.
   - We characterized the diminishing increase in *effective* Fisher information as we scaled $N_B$.

3. We observed systematic deviations from the second Bartlett identity in learned models, i.e. nominal network outputs underestimate LLR curvature. This finding highlights a critical consideration for the application of ML in high-precision statistical inference and motivated our development of a post-hoc calibration procedure.

4. We provided a comparison of different ML implementations for this parameter estimation problem; showing their respective strengths, limitations, and proposed solutions to those limitations.

This methodology is a general-purpose framework for having more precise detections of weak signals contained in a dataset. As physicists, our primary aim is to extract the maximal experimentally available information from finite datasets. The methodology introduced here provides a pragmatic route toward this goal. By treating an event collection as a single, permutation-invariant input, we can amplify extremely weak per-event signals into a bag-level statistic that is amenable to inference. Moreover, since the realized gain depends on the behavior of the ML-induced error term $\sigma_\epsilon^2(N_B)$, if $\sigma_\epsilon^2(N_B)$ is shown (theoretically or empirically) to grow sublinearly with $N_B$, then aggregation will systematically suppress ML-induced error. However, the general rules and conditions for such sublinear scaling remain an open question.

The primary objective of this paper was to perform a comparative analysis of this methodology in low-signal regime and to provide an initial characterization of its properties. Although we acknowledge the theoretical and the empirical limitations of this paper (see Appendix C.6 for a detailed discussion), the information-theoretic perspective given in this paper shows a nontrivial and counterintuitive result: Under certain conditions, aggregating instances into a set can allow an ML model to extract more information per instance than is achievable by a model that processes each instance individually.

This work opens several promising avenues for future research. We believe a deeper analysis of the machine learning models themselves is a critical, and often overlooked, component of phenomenological studies. As for future work, our main questions are as follows:

- For a given ML architecture and intrinsic data dimensionality $d$, what is the SNR threshold below which the model cannot perform optimally, and how does that threshold scale with dataset size and model capacity?

- Can we develop a rigorous theoretical or empirical framework to characterize the variance of the learned-error function $\sigma_\epsilon^2(N_B)$?

- Can we find robust training or architectural strategies that mitigate violation of the second Bartlett identity without degrading predictive performance?

- How general are ML behaviors across architectures, datasets, and physics tasks? Which behaviors are model- or problem-specific and which are universal?

- What is the theoretical information capacity of the Asimov vector $\bar{e}$, and how does this capacity depend on the aggregation operator and embedding dimension?

- How can MIL-specific architectures be designed or adapted to maximize set-level sufficiency for statistical inference tasks?

### DATA AND CODE AVAILABILITY

The anonymous repository for this paper can be found at this link: `https://github.com/aaa327OpenReview/MIL_for_HEP`

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

## A MATHEMATICAL PROOFS AND APPROXIMATIONS

### A.1 FISHER INFORMATION AND IT'S RELATION TO LOG-LIKELIHOOD RATIO

For a set, or "bag", of independent particle collision events $\mathcal{B} = \{\mathbf{x_i}\}_{i=1}^{N}$, we have the likelihood and log-likelihood as,

$$p(\mathcal{B} \mid \theta) = \prod_{i=1}^{N} p(\mathbf{x_i} \mid \theta), \qquad \ln p(\mathcal{B} \mid \theta) = \sum_{i=1}^{N} \ln p(\mathbf{x_i} \mid \theta). \tag{18}$$

By definition, the Fisher Information is

$$\mathcal{I}(\theta) = \text{Var}_\theta\left[\frac{\partial}{\partial \theta} \ln p(\mathcal{B} \mid \theta)\right]. \tag{19}$$

Since the reference point is fixed, and $\dfrac{\partial}{\partial \theta} \displaystyle\sum_{i=1}^{N} \ln p(\mathbf{x_i} \mid \theta_0) = 0$, we can add the zero term to variance of the score, $\text{Var}_\theta\left[\frac{\partial}{\partial \theta} \ln p(\mathcal{B} \mid \theta_0)\right]$, and obtain

$$\mathcal{I}(\theta) = \text{Var}_\theta\left[\frac{\partial}{\partial \theta} \sum_{i=1}^{N} \ln p(\mathbf{x_i} \mid \theta) - \frac{\partial}{\partial \theta} \sum_{i=1}^{N} \ln p(\mathbf{x_i} \mid \theta_0)\right] \tag{20}$$

$$= \text{Var}_\theta\left[\frac{\partial}{\partial \theta} \sum_{i=1}^{N} \lambda_i(\theta)\right] = \text{Var}_\theta\left[\sum_{i=1}^{N} s_i(\theta)\right], \tag{21}$$

where $\lambda_i(\theta)$ is the log-likelihood ratio of event $i$ with respect to the reference parameter point $\theta_0$, and the score $s_i$ is

$$s_i(\theta) = \frac{\partial}{\partial \theta} \lambda_i(\theta) = \frac{\partial}{\partial \theta} \ln p(\mathbf{x_i} \mid \theta). \tag{22}$$

Because the events are independent,

$$\mathcal{I}(\theta) = \text{Var}_\theta\left[\sum_{i=1}^{N} s_i(\theta)\right] = N \ \text{Var}_\theta\left[s_1(\theta)\right]. \tag{23}$$

Assuming the regularity conditions that permit the interchange of differentiation and integration hold, we can show that the expectation of the score is zero. Since $p(\mathbf{x_1} \mid \theta)$ is a probability density function, its integral over the entire domain is 1. Therefore, $\mathbb{E}_\theta[s_1(\theta)] = \frac{\partial}{\partial \theta} \int p(\mathbf{x_1} \mid \theta) \, d\mathbf{x_1} = \frac{\partial}{\partial \theta}(1) = 0$. With this result, the variance of the score simplifies to,

$$\text{Var}_\theta\big[s_1(\theta)\big] = \mathbb{E}_\theta\Big[\big(s_1(\theta) - \mathbb{E}_\theta[s_1(\theta)]\big)^2\Big] = \mathbb{E}_\theta\big[(s_1(\theta))^2\big]. \tag{24}$$

## A.2 FISHER INFORMATION APPROXIMATIONS

Consider testing $H_0 : \theta = \theta_0$ versus $H_1 : \theta = \theta_1 = \theta_0 + \Delta\theta$ where $\Delta\theta$ is small. According to Neyman-Pearson lemma, for a dataset $D$, the optimal test statistic is the LLR $\Lambda(D|\theta_1, \theta_0)$ . For small $\Delta\theta$, we can Taylor expand $\ln p(D|\theta_1)$ around $\theta_0$:

$$\ln p(D|\theta_1) \approx \ln p(D|\theta_0) + \frac{\partial \ln p(D|\theta)}{\partial \theta}\Big|_{\theta_0} \Delta\theta + \frac{1}{2}\frac{\partial^2 \ln p(D|\theta)}{\partial \theta^2}\Big|_{\theta_0} (\Delta\theta)^2 \tag{25}$$

$$\Lambda(D|\theta_1, \theta_0) \approx S_D(\theta_0)\Delta\theta + \frac{1}{2}H_D(\theta_0)(\Delta\theta)^2 \tag{26}$$

where $S_D(\theta_0)$ is the score and $H_D(\theta_0)$ is the Hessian (second derivative) for the full dataset. The $\mathbb{E}[S_D(\theta_0)|\theta_0] = 0$ under $H_0$, and thanks to the second Bartlett Identity we have $\mathbb{E}[\partial^2 \ln p/\partial\theta^2] = -\mathbb{E}[(\partial \ln p/\partial\theta)^2] = -I(\theta)$. Therefore, if we take the expectation of $\Lambda$ under $H_0$:

$$\mathbb{E}[\Lambda|\theta_0] \approx \mathbb{E}[S_D(\theta_0)|\theta_0]\Delta\theta + \frac{1}{2}\mathbb{E}[H_D(\theta_0)|\theta_0](\Delta\theta)^2 \tag{27}$$

$$= 0 + \frac{1}{2}(-I(\theta_0))(\Delta\theta)^2 = -\frac{1}{2}I(\theta_0)(\Delta\theta)^2 \tag{28}$$

By definition, the Fisher Information is $I(\theta_0) = \mathbb{E}[S_D(\theta_0)^2|\theta_0]$. To find the variance, we first approximate the expectation of the squared LLR. By retaining only the lowest-order term in $\Delta\theta$ , we have:

$$\Lambda^2 \approx \left(S_D(\theta_0)\Delta\theta + \frac{1}{2}H_D(\theta_0)(\Delta\theta)^2\right)^2 \approx S_D(\theta_0)^2(\Delta\theta)^2 \tag{29}$$

The variance of $\Lambda$ under $H_0$ is therefore:

$$\text{Var}[\Lambda|\theta_0] = \underbrace{\mathbb{E}[\Lambda^2|\theta_0]}_{\approx I(\theta_0)(\Delta\theta)^2} - \underbrace{(\mathbb{E}[\Lambda|\theta_0])^2}_{\mathcal{O}((\Delta\theta)^4)} \tag{30}$$

$$\text{Var}[\Lambda|\theta_0] \approx I(\theta_0)(\Delta\theta)^2 \tag{31}$$

Therefore, under $H_0$, the LLR distribution has mean $\mathbb{E}[\Lambda|\theta_0] \approx -I(\theta_0)(\Delta\theta)^2/2$ and variance $\text{Var}[\Lambda|\theta_0] \approx I(\theta_0)(\Delta\theta)^2$. Since Fisher Information is locally constant for small $\Delta\theta$ (because $\Lambda$ is asymptotically $\chi^2$ distributed, $I(\theta_1) \approx I(\theta_0)$ ), through similar calculations shown above, one can show that the mean $\mathbb{E}[\Lambda|\theta_1] \approx +I(\theta_0)(\Delta\theta)^2/2$ and the variance $\text{Var}[\Lambda|\theta_1] \approx I(\theta_0)(\Delta\theta)^2$ under the $H_1$ hypothesis.

# B IMPLEMENTATION

## B.1 DATA GENERATION AND FEATURE SELECTION

The dataset used for this research is hadron-level high-energy collision events created by Monte Carlo simulations using `MadGraph5_aMC@NLO` (v3.6.2) (Frederix et al., 2021) interfaced with the `SMEFTsim` (v3.0) UFO model (Brivio et al., 2017) to incorporate EFT effects. We have generated $10^6$ collision events for each parameter value in the set of $c_{HW}$ values. We choose the $c_{HW}$ values to be in the range of $[-10, 10]$ with increments of $\pm 1.0$, and in the range of $[-0.9, 0.9]$ with increments of $\pm 0.1$, resulting in a total of 39 discrete values.

The analysis focuses on a specific signal process sensitive to the $c_{HW}$ parameter and a corresponding background process chosen for its similar kinematic signature:

**Signal process:** Vector Boson Fusion (VBF) production of a Higgs boson, which subsequently decays via $H \to WW \to \ell\nu\ell\nu$. The `MadGraph5` command used is:

```
import model SMEFTsim_top_MwScheme_UFO-massless
generate u d > u d h $$ w+ w- / z a QCD=0 NP=1 NPcHW=1,
  h > e+ ve e- ve~ / z QCD=0 NP=1 NPcHW=1
```

**Background process:** We chose a kinematically similar irreducible process, a VBF production of a di-boson (ZZ) pair, with one Z decaying leptonically ($Z \to \ell\ell$) and the other invisibly ($Z \to \nu\nu$). The generation is done with no EFT effects ($c_{HW} = 0$). The `MadGraph5` command used is:

```
import model SMEFTsim_top_MwScheme_UFO-massless
define vl = ve vm vt
define vl~ = ve~ vm~ vt~
generate u d > u d z z QCD=0 NP=0 NPcHW=0, (z > e+ e-),
(z > vl vl~) QCD=0 NP=0 NPcHW=0
```

Both processes result in the same final state signature of two forward jets, two charged leptons, and significant missing transverse energy, making them an ideal test case for a method designed to distinguish between hypotheses based on subtle kinematic differences.

The `run_param.dat` parameter card file was modified for each run to set the specific value of $c_{HW}$ while keeping all other Wilson coefficients at their Standard Model value of zero.

The features used for model training are detailed in Table 1. They include both low-level four-vector components for the final state particles and a set of high-level, physically-motivated engineered variables.

Table 1: Features included in the training dataset. The features are categorized into low-level kinematic variables and high-level engineered features. For pairs of particles, the indices 0 and 1 (e.g., $\ell_0, \ell_1$) refer to the leading and subleading particles sorted by transverse momentum ($p_T$), respectively.

| Feature Name | Description | Mathematical Definition |
|---|---|---|
| **Low-Level Features** | | |
| $p_{T,i}, \eta_i, \phi_i, E_i, m_i$ | Basic kinematic properties (transverse momentum, pseudorapidity, azimuthal angle, energy, and mass) for each particle $i \in \{\ell_0, \ell_1, q_0, q_1\}$. | - |
| $E_T^{\text{miss}}, \phi^{\text{miss}}$ | Missing transverse energy and its azimuthal angle, defined from the negative vector sum of all visible transverse momenta. | $\vec{p}_T^{\text{miss}} = -\sum_k \vec{p}_{T,k}^{\text{vis}}$ |
| Particle ID | One-hot encoded flags indicating the type or charge of final state particles (e.g., electron vs. positron, down-type vs. up-type quark). | - |
| **High-Level (Engineered) Features** | | |
| $m_{\ell\ell}, m_{qq}$ | Invariant mass of the di-lepton or di-quark system. | $\sqrt{E_{\text{sys}}^2 - |\vec{p}_{\text{sys}}|^2}$ |
| $p_{T,\ell\ell}, p_{T,qq}$ | Transverse momentum of the di-lepton or di-quark system. | $|\vec{p}_{T,1} + \vec{p}_{T,2}|$ |
| $\Delta\phi_{\ell\ell}, \Delta\phi_{qq}$ | Signed difference in $\phi$ between the two particles, with the sign determined by their ordering in $\eta$. | $(\phi_a - \phi_b)$ s.t. $\eta_a > \eta_b$ |

## B.2 MACHINE LEARNING PIPELINE

To ensure a fair comparison and robust conclusions, a consistent training pipeline was used for all models unless otherwise specified. The pipeline was implemented in `TensorFlow` (Abadi et al., 2015) and experiment tracking was managed with `wandb` (Biewald, 2020).

### B.2.1 NEURAL NETWORK MODEL ARCHITECTURE

The core architecture is a simple Multi-Layer Perceptron (MLP) with 11,201 trainable parameters, chosen deliberately to demonstrate that the performance gains stem from the set-based aggregation method rather than from architectural complexity. The network consists of:

1. A normalization layer, adapted to the training data.

2. Three fully-connected hidden layers with 64 neurons each. Each layer uses the ELU activation function (Clevert et al., 2016), Batch Normalization (Ioffe & Szegedy, 2015), and is regularized with Dropout (rate=0.1) (Srivastava et al., 2014) and an L2 kernel regularizer $(10^{-3})$.

3. A global average pooling layer operates across the "events-in-bag" axis of the output embeddings from the final hidden layer. This produces a single, fixed-size summary vector for the entire bag, which we call the *Asimov Vector*.

4. A final output is a single neuron with a sigmoid (for binary) or softmax (for multi-class) activation function.

### B.2.2 DATA HANDLING AND TRAINING PROCEDURE

The dataset was first partitioned at the event level to prevent data leakage: 20% was held out as a final test set, with the remainder is then shuffled with the experiments seed value and split into training (80%) and validation (20%) sets.

- **Dynamic bags:** As a simple data augmentation method we have created dynamic bags. At the beginning of each training epoch, the events within the training set are randomly shuffled and re-grouped into new, unique bags. Although it was essential for multi-class classifiers to have for stable LLR profile predictions, dynamic bags did not have any meaningful effect on performance for binary classifiers and PNNs in our case study.

- **Training and optimization:** Depending on the problem, binary cross entropy or categorical cross entropy is used as the loss function. The models were trained using the Adam optimizer, with an initial learning rate of $10^{-3}$, and reducing the learning rate up to $10^{-4}$, if no improvements were seen for a predetermined `PATIENCE` number of epochs. Early stopping is applied if there is no improvement after `2*PATIENCE` epochs after the last learning rate reduction, restoring the model weights from the epoch with the best validation loss. Validation loss was chosen as the monitor to determine the early stopping and learning rate reduction point.

- **Batching strategy:** To maintain a consistent number of gradient updates per epoch across experiments with different bag sizes $(N_B)$, the batch size was set dynamically as `floor(80000 / N_B)`. This provides a stable basis for comparing the training dynamics.

We tracked all of the training runs and made sure that no model is stopped before reaching its performance plateau.

### B.2.3 BINARY CLASSIFICATION

The binary classification task was designed to test the model's fundamental ability to distinguish between two competing hypotheses in a low-signal environment. We define the null hypothesis, $H_0$, as the Standard Model process and the alternative hypothesis, $H_1$, as the SMEFT process with a specific, non-zero Wilson coefficient $(c_{HW} \neq 0)$.

The training dataset was constructed from "bags" of events. A bag was labeled 1 (positive class) if its signal events were drawn from the SMEFT signal sample $(H_1)$. Conversely, a bag was labeled 0

(negative class) if its signal events were drawn from the corresponding SM signal sample ($H_0$). The model was then trained using a binary cross-entropy loss function to distinguish between these two categories of bags based on their aggregated kinematic information.

**XGBoost training**  We trained a hyperparameter-optimized XGBoost baseline for binary classification. The hyperparameters were optimized with the sophisticated framework Optuna (Akiba et al., 2019) over a search space including `n_estimators`, `max_depth`, `learning_rate`, `min_child_weight`, `subsample`, and `reg_lambda` using stratified 3-fold cross-validation and 50 Optuna trials. For the task, we set `objective='binary:logistic'` for the model training, and we optimized AUC ("roc_auc") in the hyperparameter search. We used the histogram tree method for stability, and the best hyperparameters were refit on the training data and evaluated on the held-out test set.

### B.2.4    MULTI-CLASS CLASSIFICATION

To investigate the model's capability for parameter estimation, we framed the problem as a multi-class classification task. The goal is to identify the correct parameter value, $\theta$, for a given bag of events from a discrete set of $K$ possible hypotheses, $\theta_1, \theta_2, ..., \theta_K$.

For this setup, a bag of events $\{\mathbf{x_i}\}_{i=1}^{N_B}$ where all events are Monte Carlo sampled from the distribution $p(\mathbf{x_i}|\theta_k)$ is assigned the integer class label $k$. During training, these integer labels are converted into a one-hot encoded vector of length $K$. For example, a bag corresponding to the third hypothesis, $\theta_3$, would be given the label `[0, 0, 1, 0, ..., 0]`. For our analysis, we trained the model with the $\theta_k$ taking a value in the range of $[-1, 1]$ with increments of $\pm 0.1$.

The neural network's final layer is equipped with a softmax activation function producing $K$ output nodes, corresponding to the probability of the bag belonging to each class. The model is then trained to minimize the categorical cross-entropy loss between its prediction and the true one-hot encoded label.

### B.2.5    PARAMETERIZED NEURAL NETWORKS

The Parameterized Neural Network (PNN) approach was investigated as an alternative method for parameter estimation. Unlike the multi-class classifier which assigns a bag to one of several discrete classes, the PNN is designed to learn a continuous functional relationship between the event kinematics $\mathbf{x}$, and the parameter of interest $\theta$.

The training data for the PNN was structured as a set of labeled pairs. Each input sample given to the network consisted of both a bag of kinematic events and a single candidate value for the parameter $c_{HW}$. The model's objective was framed as a binary classification task: to predict whether the kinematics in the bag are consistent with the paired $c_{HW}$ value.

To achieve this, the training dataset was composed of:

- **Positive examples (label = 1):** A bag of events generated with a specific Wilson coefficient, $\theta_k$, is paired with its true parameter value. The input is thus a tuple: $(\mathcal{B}_{c_{HW}=k}, \theta_{c_{HW}=k})$.

- **Negative examples (label = 0):** Two types of bags are generated: in one case, the bag of events generated under the SM hypothesis ($\theta_{SM} = 0$) is deliberately paired with a false, non-zero Wilson coefficient, $\theta_k$; in the other case, the bag of events generated under the SMEFT hypothesis ($\theta_{SMEFT} \neq 0$) is paired with $\theta_k = 0$. The inputs are the tuples: $(\mathcal{B}_{c_{HW}=0}, \theta_{c_{HW}\neq0})$ or $(\mathcal{B}_{c_{HW}\neq0}, \theta_{c_{HW}=0})$.

By training on a balanced set of these positive and negative examples with a binary cross-entropy loss, the network learns a function $f(x, \theta)$ that approximates the likelihood ratio. After training, this function can be used for inference: for a given bag of data events, the parameter $\theta$ can be scanned over a continuous range. The value of $\theta$ that maximizes the network's output is taken as the maximum likelihood estimate for that bag, and the full scan of the output produces the profile of the LLR.

## C DETAILED REPORT ON EXPERIMENTAL RESULTS AND PROCEDURES

This section provides a comprehensive report on our analysis, with supplementary plots and discussions.

### C.1 ON THE INTERPRETATION OF MODEL DECISIONS

In our analysis, we observed that the ML models, in their effort to minimize the global loss function, can adopt decision strategies that are locally counterintuitive. Because the optimization objective is the overall loss across all examples, the model may learn to accept a higher loss for certain types of events or certain classes (e.g., at low $c_{HW}$) in exchange for a much larger gain on other, more easily separable examples contained in the training dataset.

This behavior is evident in the box plots of the probability predictions (Figures 10 and 18) and in the plots of the individual predictions (Figures 11 and 19). Particularly for the event-level case ($N_B = 1$), the model does not express high confidence at the true SM value ($c_{HW} = 0$). Instead, the highest average predictions are often assigned to the most extreme $c_{HW}$ values at the edge of the training range. We interpret this not as a simple failure, but as an emergent strategy. Since the kinematic differences are largest at these extreme points, the model can achieve the lowest loss by confidently identifying them. The resulting output is not a "probability" in the classic sense, but an emergent probability distribution prediction strategy for aggregate evaluation metrics.

This underscores a critical point: one cannot naively interpret the nominal output of a classifier as a true posterior probability without careful validation. As we demonstrate with the non-smooth profile of the LLR values of multi-class classifiers (Section C.4) and the unphysical predictions of PNNs (Section C.5), ML models will exploit any asymmetry or feature in the training setup to minimize their objective, leading to powerful but sometimes unintuitive results.

### C.2 JUSTIFICATION OF THE ESTIMATOR CORRECTION AND INFORMATION MEASUREMENT

Our investigation revealed that the raw Maximum Likelihood Estimate (MLE), $\hat{\theta}$, derived from the ML models exhibits two non-ideal behaviors:

1. The LLR curvature does not match the MLE variance ($I_{\text{curv}} \neq I_{\text{MLE}}$), violating the second Bartlett identity.

2. The models are not unbiased estimators, i.e. $\mathbb{E}[\hat{\theta}] \neq \theta_{\text{true}}$.

This section details the procedures used to correct for these effects and justifies why our primary measurement of the effective Fisher Information remains sound.

#### C.2.1 PROCEDURE 1: LLR CURVATURE CALIBRATION

As it was explained in Section 4.2, and can be seen in Tables 2 and 3, the nominal predictions of the ML models systematically violate the second Bartlett identity. To construct confidence intervals with correct frequentist coverage, we apply a post-hoc calibration by introducing a confidence interval calibration constant, $c_{\text{cicc}}$, which serves to rescale the LLR values: $\hat{\Lambda}_{\text{calib}}(\theta) = c_{\text{cicc}} \cdot \hat{\Lambda}(\theta)$.

The Maximum Likelihood Estimate (MLE) point, $\hat{\theta}$, is the parameter value that minimizes $\hat{\Lambda}(\theta)$. Since $c_{\text{cicc}}$ is a positive constant, the value of $\theta$ that minimizes $\hat{\Lambda}(\theta)$ is the exact same value that minimizes $\hat{\Lambda}_{\text{calib}}(\theta)$. Therefore, the MLE is invariant under calibration, and the Fisher Information calculated from MLE is also invariant under such calibration. For test statistics $T(D) = \sum_{j=1}^{M} \hat{\Lambda}_j$, we have,

$$I_{\text{MLE}}(T) \equiv \frac{1}{\text{Var}(\hat{\theta}(T))} \tag{32}$$

But Fisher Information calculated from the curvature of the parabolic fit scales linearly with the calibration constant:

$$I_{\text{curv}}(T_{\text{calib}}) = \mathbb{E}\left[-\frac{d^2}{d\theta^2}(c_{\text{cicc}} \cdot T)\right] = c_{\text{cicc}} \cdot \mathbb{E}\left[-\frac{d^2 T}{d\theta^2}\right] = c_{\text{cicc}} \cdot I_{\text{curv}}(T) \tag{33}$$

By enforcing the Bartlett identity on our calibrated result (i.e., setting $I_{\text{curv}}(T_{\text{calib}}) = I_{\text{MLE}}(T)$), we can *empirically determine* the necessary correction factor:

$$c_{\text{cicc}} = \frac{I_{\text{MLE}}(T)}{I_{\text{curv}}(T)} = \frac{1/\operatorname{Var}(\hat{\theta}(T))}{\mathbb{E}\left[-\frac{d^2 T}{d\theta^2}\right]} \tag{34}$$

This procedure allows us to use the empirically measured $I_{\text{MLE}}$ as our robust proxy for the effective Fisher Information ($I_{\text{eff}}$), as it correctly encapsulates all effects on the estimator's variance, while the curvature is separately corrected to ensure valid confidence intervals. Our goal is to measure how this quantity, $I_{\text{eff}} \approx I_{\text{MLE}}$, scales with the bag size $N_B$.

### C.2.2 PROCEDURE 2: POST-HOC BIAS CORRECTION

To report an unbiased central value for $\hat{\theta}$ and to validate our calibrated confidence intervals, we applied a mathematically justified and rigorous post-hoc correction for the observed bias.

The models exhibited a small but consistent bias, defined as $b(\hat{\theta}) = \mathbb{E}[\hat{\theta}] - \theta_{\text{true}}$. Since our null hypothesis is centered at $\theta_{\text{true}} = 0$, this simplifies to $b(\hat{\theta}) = \mathbb{E}[\hat{\theta}]$. For a set of $N$ number of MLEs $(\hat{\theta}_1, ..., \hat{\theta}_N)$ from the pseudo-experiments, we first estimate the bias as the sample mean, $\hat{b} = \frac{1}{N}\sum_i \hat{\theta}_i$. By the Law of Large Numbers, this sample mean is a consistent estimator of the true bias. We then define the corrected estimate, $\hat{\theta}'$, as:

$$\hat{\theta}'_i = \hat{\theta}_i - \hat{b} \tag{35}$$

The validity of using $I_{\text{MLE}} = 1/\operatorname{Var}(\hat{\theta}')$ as our sensitivity measure, even after this correction, is justified by its negligible impact on variance. Since the variance of the corrected estimator, $\operatorname{Var}(\hat{\theta}')$, is related to the variance of the original estimator, $\operatorname{Var}(\hat{\theta})$, by the standard relation for deviations from a sample mean, we have the relation:

$$\operatorname{Var}(\hat{\theta}') = \operatorname{Var}\left(\hat{\theta} - \frac{1}{N}\sum_{i=1}^{N}\hat{\theta}_i\right) = \operatorname{Var}(\hat{\theta})\left(1 - \frac{1}{N}\right) \tag{36}$$

In our analysis, we constructed 200 confidence intervals. Since $N = 200$, the bias correction changes the variance only about $0.5\%$, which is a negligible effect. Furthermore, this bias correction procedure is applied for all bag sizes, therefore its overall affect on the scaling behavior of the ML models with respect to bag size is much more minuscule. Therefore this minor and consistent procedure does not affect the study of the overall scaling behavior of $I_{\text{MLE}} \approx I_{\text{eff}}$ with respect to bag size, and it ensures that our corrected estimator $\hat{\theta}'$ is asymptotically unbiased at the null hypothesis, as required for proper frequentist coverage testing.

### C.3 BINARY CLASSIFIERS

As explained in Section 4.1, we trained five MLP models at each bag size and background contamination level to study robustness and analyze variations in performance in different training runs. In high-energy physics, creating pure signal samples is often infeasible due to irreducible background processes. Therefore we need to understand if, when, and how the ML model performance degrades. Although a comprehensive study of background effects is beyond the scope of this work, we performed a targeted study to test the classifier's robustness to noise and determine if it could behave as an ideal discriminator.

As demonstrated by Nachman & Thaler (2021), artificial bag-level predictions can be obtained from single-instance predictors by composing per-event likelihood contributions. Since events are i.i.d., the joint probability of a set of events $\{\mathbf{x_i}\}_{i=1}^{N}$ under a model parameterized by $\theta$ is the product of individual event probabilities:

$$p(\{\mathbf{x_i}\}_{i=1}^{N}|\theta) = \prod_{i=1}^{N} p(\mathbf{x_i}|\theta). \tag{37}$$

This likelihood-based approach can be implemented in a numerically stable manner by summing the per-event logits and mapping the result to a score with the sigmoid function.

$$p(\{\mathbf{x_i}\}_{i=1}^N | \theta) = \frac{1}{1 + \exp\left(- \sum_{i=1}^N \log\left(\frac{p_i}{1-p_i}\right)\right)} \tag{38}$$

Figures 6 and 7 compare single-instance MLP and XGBoost baselines against multiple-instance MLP models. MIL's resilience to performance degradation at low SNR levels provides a strong evidence for the theoretical predictions stated in Section 3.

**On the traditional histogram-based analysis** In the standard high-energy physics analysis paradigm, the nominal output of a classifier is not directly interpreted as a true event probability. Therefore the ML models are often employed as a dimensionality reduction tool. Its function is to map the high-dimensional feature vector of an event to a single discriminant value. The histogram of this classifier output value is then taken for both signal (BSM) and background (SM) simulations to create shape templates. The final physics measurement is extracted via a binned maximum likelihood fit that compares these templates to the distribution observed in the data.

The statistical power of this entire procedure is contingent upon a discernible separation between the signal and background histogram shapes. In the low-signal regime studied here, per-event classifier outputs produce nearly overlapping histograms (Figure 4a), leaving little shape information for a fit to exploit.

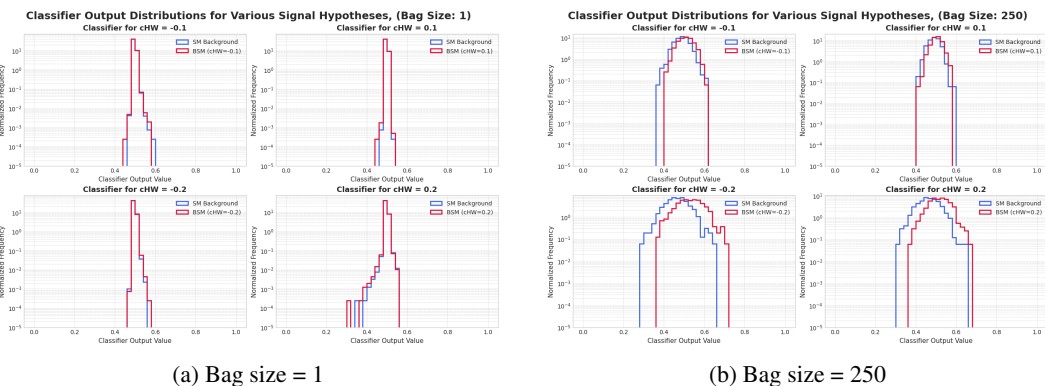

(a) Bag size = 1                    (b) Bag size = 250

Figure 4: Distributions of the ensemble classifier output for event-by-event ($N_B = 1$, left) and set-based ($N_B = 250$, right) classification. Larger versions of these plots are shown in Figures 8 and 9.

**Parameter estimation with binary classifiers** To extend the binary classification framework towards parameter estimation, we attempted to construct a continuous LLR profile from our discrete set of classifiers. For each $c_{HW}$ value in the dataset (see Appendix B.1), an ensemble prediction was first generated by averaging five independently trained models. Despite these measures, this approach proved to be unstable, even at very large bag sizes. Since each binary classifier is trained in isolation, there is no enforcement of continuity between adjacent $c_{HW}$ points. This independence resulted in extremely noisy LLR profiles unsuitable for robust confidence interval calculation, motivating the transition to the inherently coupled prediction frameworks of multi-class classifiers and PNNs.

### C.4  MULTI-CLASS CLASSIFIERS

The discrete binary classification approach produces a test statistic from a set of independently trained models. In order to couple the predictions for the *all* $c_{HW}$ values which are analyzed for the confidence interval calculations, we transitioned to a multi-class framework. Since the softmax activation function is used in the final layer, the model is forced to learn the relative importance of each hypothesis $\theta_k$, as the output probabilities must sum up to one.

However, our straightforward implementation presents a challenge. The model treats each one-hot encoded $\theta_k$ value as an independent category and has no "inductive bias" that informs it of the ordinal relationship between the classes (e.g., that $c_{HW} = 0.1$ is next to $c_{HW} = 0.2$) or that the resulting LLR profile should be locally parabolic. In the extremely low-signal regime of our study, this makes it difficult for the network to learn a *smooth* function of $\theta$. It is possible to design an ML architecture where such an inductive bias is enforced, for instance, by constraining the output to follow a specific functional form, but that is left for future work.

As illustrated in Figure 5, this lack of inductive bias manifested as individual models producing non-smooth LLR profiles, particularly for small bag sizes ($N_B \leq 10$). To mitigate this instability, we created an ensemble model for each bag size by averaging the predictions of 20 models that were trained on the same hyperparameters but with different initialization seed values. This ensembling technique proved highly effective, producing the relatively stable and physically plausible LLR profiles required for parameter estimation. For both multi-class classifiers and PNNs, we observed that the ensemble models were much more stable in terms of both their predictions, and the variations in the Fisher information. (see Figures 2 and 3a)

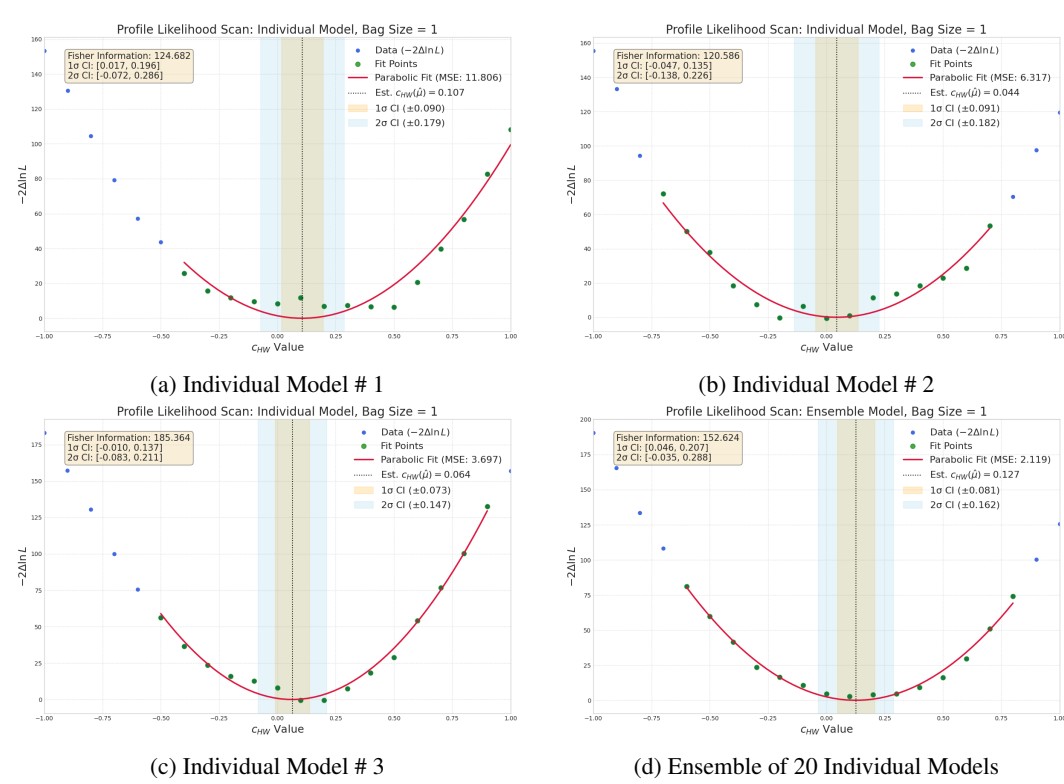

Figure 5: Multi-class classifier: LLR values, and the parabolic fits for the same 1000-event pseudo-experiment.

The final results for the ensemble multi-class models are summarized in Table 2. From left to right, the columns of the table are: the bag size, whether it is calibrated or uncalibrated, the confidence interval calibration constant ($c_{\text{cicc}}$), the percentage of the $1\sigma$ confidence interval that covers the true value, mean fisher information across all the confidence interval calculations, the bias constant $\hat{B}$ (in $\theta$ values, and as it's defined in equation 35), and the Mean Square Error (MSE) between the parabola fit and its fit values (i.e. the $-2\hat{\Lambda}$'s).

**Notes on the Analysis Procedure**   A few details are pertinent to the interpretation of Table 2. First, the coverage is calculated from 200 pseudo-experiments, meaning its statistical precision is limited to $\pm 0.5\%$. Second, the MSE of the fit naturally increases after calibration, as the $c_{\text{cicc}}$ factor scales up the LLR values and thus the absolute deviations from the parabolic fit. Furthermore, the LLR is *locally parabolic* near the maximum likelihood estimate point. Therefore to ensure a robust

parabolic fit to the LLR profile, we used a small fit window of $\pm 0.4$ $c_{HW}$ around the minimum $-2\hat{\Lambda}$ point. For the $N_B = 1$ case, which exhibited much higher MSE, this window was expanded to $\pm 0.7$ $c_{HW}$ to make the fit less susceptible to local fluctuations, resulting in a much more stable information measurement. Figures 14 and 15 show two examples of confidence interval calculations to validate our approach.

Table 2: The results for the ensemble model of **multi-class classifiers**.

| Bag Size | Calibration | CI Const. ($c_{cicc}$) | Coverage (%) | Mean Fisher Info. | Bias ($\theta$) | MSE ($\Lambda$) |
|---|---|---|---|---|---|---|
| 1 | Uncalibrated | 1.0 | 75.5 | 164.6 | 0.098 | 1.939 |
|   | Calibrated | 1.329 | 68.5 | 218.8 | 0.098 | 3.426 |
| 10 | Uncalibrated | 1.0 | 75.0 | 209.0 | 0.007 | 0.073 |
|   | Calibrated | 1.316 | 68.5 | 275.0 | 0.007 | 0.126 |
| 20 | Uncalibrated | 1.0 | 74.5 | 200.2 | 0.009 | 0.063 |
|   | Calibrated | 1.363 | 68.5 | 272.7 | 0.009 | 0.116 |
| 25 | Uncalibrated | 1.0 | 76.0 | 200.8 | 0.014 | 0.085 |
|   | Calibrated | 1.360 | 68.5 | 273.2 | 0.014 | 0.158 |
| 50 | Uncalibrated | 1.0 | 77.5 | 190.4 | 0.021 | 0.029 |
|   | Calibrated | 1.531 | 68.5 | 291.5 | 0.021 | 0.068 |
| 100 | Uncalibrated | 1.0 | 81.0 | 151.1 | 0.025 | 0.014 |
|   | Calibrated | 1.956 | 68.5 | 295.5 | 0.025 | 0.055 |
| 125 | Uncalibrated | 1.0 | 82.0 | 137.7 | 0.020 | 0.010 |
|   | Calibrated | 2.214 | 68.5 | 304.9 | 0.020 | 0.050 |
| 200 | Uncalibrated | 1.0 | 89.0 | 101.8 | 0.029 | 0.006 |
|   | Calibrated | 2.988 | 68.5 | 304.1 | 0.029 | 0.052 |
| 250 | Uncalibrated | 1.0 | 91.5 | 80.4 | 0.024 | 0.004 |
|   | Calibrated | 3.577 | 68.5 | 287.7 | 0.024 | 0.045 |

## C.5 PARAMETERIZED NEURAL NETWORKS

Our investigation of the Parameterized Neural Network approach for this high-precision task revealed significant instabilities. We identified two primary, interconnected challenges: the unphysical nature of the LLR profile and an extreme sensitivity to the symmetry of the training data.

The first issue stems from the PNN's unnormalized output. Unlike a multi-class softmax, the PNN's outputs for different $\theta$ values are independent, making the absolute scale of the predicted probabilities arbitrary. This means that a simple rescaling of the output can drastically alter the resulting confidence interval, rendering the nominal LLR profile unreliable. We attempted a post-hoc correction, specifically by normalizing the probability outputs by their sum over the $c_{HW}$ analysis range, but it did not produce a stable or improved LLR profile, confirming that simple post-hoc rescaling is insufficient to solve the problem.

The second, more fundamental issue is the PNN's sensitivity to training data asymmetries. The training scheme, which pairs kinematic bags with $\theta$ values, effectively asks the model to solve many independent binary classification tasks simultaneously, i.e. having a single model to take the place of the discrete binary classifiers for all $\theta_{c_{HW}}$ values, as discussed in Section C.3. We found that this makes the model highly susceptible to learning and exploiting any imbalance in how the "true match" (positive) and "false match" (negative) examples are constructed.

To address the model's sensitivity to these training asymmetries, we systematically explored several training configurations. We found that simpler, asymmetric schemes consistently led to critical failure modes, such as extrapolation failure at the SM point or the memorization of a direct mapping from the $\theta$ feature to the label, disregarding the kinematic data.

Therefore, the analysis presented in this paper was performed using a fully symmetric training set, constructed with both positive (($\mathcal{B}_{c_{HW}=k}, \theta_{c_{HW}=k}$)) and negative examples (($\mathcal{B}_{c_{HW}=0}, \theta_{c_{HW}\neq0}$) or ($\mathcal{B}_{c_{HW}\neq0}, \theta_{c_{HW}=0}$)) for all $\theta_k$. The negative examples for the $\theta_{c_{HW}=0}$ hypothesis were created

from a mixture of kinematics with $c_{HW}$ values near zero: 20% each from $c_{HW} = \pm 0.2$, 30% each from $c_{HW} = \pm 0.1$; totaling enough bags to have $10^6$ events for the negative samples, same as its positive counterpart. Despite this principled construction, a subtle but critical imbalance remained. Since most of the SM kinematics ($\mathcal{B}_{c_{HW}=0}$) the model sees in training are negative examples with label 0, the model learned this strong correlation. This, in turn, caused it to assign decreasingly low probabilities as it became more certain of the SM kinematics (Figure 18), resulting in incorrect predictions at the reference point (Figure 20).

The detailed numerical results for the ensemble PNN models are presented in Table 3. Unlike the multi-class classifiers, PNNs always have smooth profile LLRs. Therefore, for all bag sizes we set the fit range of the parabolic curve to be the constant value of $\pm 0.4\ c_{HW}$ from the minimum $-2\hat{\Lambda}$ point.

Table 3: The results for the ensemble model of **Parameterized Neural Networks**.

| Bag Size | Calibration | CI Const. ($c_{cicc}$) | Coverage (%) | Mean Fisher Info. | Bias ($\theta$) | MSE ($\Lambda$) |
|---|---|---|---|---|---|---|
| 1 | Uncalibrated | 1.0 | 76.5 | 175.9 | 0.031 | 0.386 |
| | Calibrated | 1.325 | 68.5 | 233.2 | 0.031 | 0.679 |
| 10 | Uncalibrated | 1.0 | 75.0 | 174.2 | 0.034 | 0.018 |
| | Calibrated | 1.283 | 68.5 | 223.4 | 0.034 | 0.029 |
| 20 | Uncalibrated | 1.0 | 77.0 | 170.5 | 0.028 | 0.015 |
| | Calibrated | 1.331 | 68.5 | 226.9 | 0.028 | 0.027 |
| 25 | Uncalibrated | 1.0 | 77.5 | 165.2 | 0.028 | 0.013 |
| | Calibrated | 1.336 | 68.5 | 220.8 | 0.028 | 0.023 |
| 50 | Uncalibrated | 1.0 | 83.0 | 126.9 | 0.027 | 0.008 |
| | Calibrated | 1.614 | 68.5 | 204.9 | 0.027 | 0.020 |
| 100 | Uncalibrated | 1.0 | 89.5 | 94.0 | 0.039 | 0.006 |
| | Calibrated | 2.015 | 68.5 | 189.4 | 0.039 | 0.024 |
| 125 | Uncalibrated | 1.0 | 92.0 | 74.7 | 0.040 | 0.003 |
| | Calibrated | 2.480 | 68.5 | 185.1 | 0.040 | 0.021 |
| 200 | Uncalibrated | 1.0 | 100.0 | 33.9 | 0.054 | 0.000 |
| | Calibrated | 7.504 | 68.5 | 254.4 | 0.054 | 0.022 |
| 250 | Uncalibrated | 1.0 | 100.0 | 23.1 | 0.069 | 0.000 |
| | Calibrated | 11.982 | 68.5 | 276.9 | 0.069 | 0.016 |

## C.6 FINAL REMARKS

In this work, we analyzed the behavior of several ML estimators on a simplified model for a parameter estimation problem. Below, we summarize the main theoretical and empirical limitations and clarify which aspects remain open for future study.

As stated in Section 2, the datasets used in our experiments are simplified relative to real LHC data. The signal-to-background ratios used here are simplified relative to the real LHC data. Detector effects (e.g. pile-up and correlated detector responses) can violate the i.i.d. assumptions; thus instance dependencies must be addressed before applying our pipeline to full experimental data.

Moreover, our detailed analysis confirms that while the set-based ML estimators are powerful, they are not "ideal" statistical tools out of the box. We identified several important behaviors that warrant further investigation. The nominal, per-bag predictions can be unphysical, only becoming meaningful when aggregated into a full test statistic. More fundamentally, we found a systematic violation of the second Bartlett identity, requiring a calibration ($c_{cicc}$) to ensure correct frequentist coverage.

Likewise, we calculated the confidence intervals, Maximum-Likelihood Estimate (MLE) point, and the resulting Fisher Information metric through parabolic curve fitting. However, the error propagation, i.e. the theoretical and empirical uncertainty induced by this fitting, as well as the effect of the post-hoc bias correction procedure stated in the Appendix C.2 was not rigorously derived in this work. Additionally, our analytic approximations for effective Fisher Information used first-order ex-

pansions. Higher-order corrections and heteroscedastic effects were not fully explored; adjustments for heteroscedastic variance are necessary for the general case.

Furthermore, when the bag size $N_B$ becomes large, the number of available bags $M$ for training and testing necessarily decreases. Consequently, when $N_B$ is large the number of independent bags $M$ shrinks, and averages such as $\frac{1}{M} \sum_j \epsilon_j$ may not approximate their expectation reliably; this weakens asymptotic guarantees and complicates bias correction.

These findings motivate a clear research agenda that focuses on refining this methodology. Future work should focus on developing methods to mitigate these observed effects, for instance, by designing novel loss functions or regularization terms that enforce the Bartlett identity during training, or by creating architectures specifically designed to minimize parameter-dependent bias. A rigorous characterization of the ML error term, $\sigma_\epsilon^2(N_B)$, also remains a critical open question. While our illustrative ansatz, $\sigma_\epsilon^2(N_B) \propto C_1 \cdot \sqrt{N_B}$, is consistent with our observations, a more complete model is needed. For example, if the error contains an additional linear component ($\sigma_\epsilon^2(N_B) \propto C_1 \cdot \sqrt{N_B} + C_2 \cdot N_B$), the model could never reach the true Fisher Information. Since both of these ansatz solutions would have similar scaling behavior with respect to bag size, without proving that the $\sigma_\epsilon^2(N_B)$ is a sublinear function of $N_B$, a simple empirical analysis would not be enough to determine whether we have reached the theoretical maximum Fisher Information for a given dataset.

As we have mentioned, the primary objective of this paper was to characterize this methodology in low-signal regime, document its empirical behavior, and identify concrete failure modes. We conclude not that ML models *will* universally attain theoretical efficiency, but that it is *possible* to approach it *if* the required conditions hold. By understanding and modeling the asymptotic behavior of machine learning components, a principled analysis can be created that closes the gap between the effective information extracted and the true information latent in a dataset.

## D  ADDITIONAL PLOTS.

For binary-classification tasks, results are presented for individual models; for multi-class classification and parameterized networks, results correspond to ensemble models.

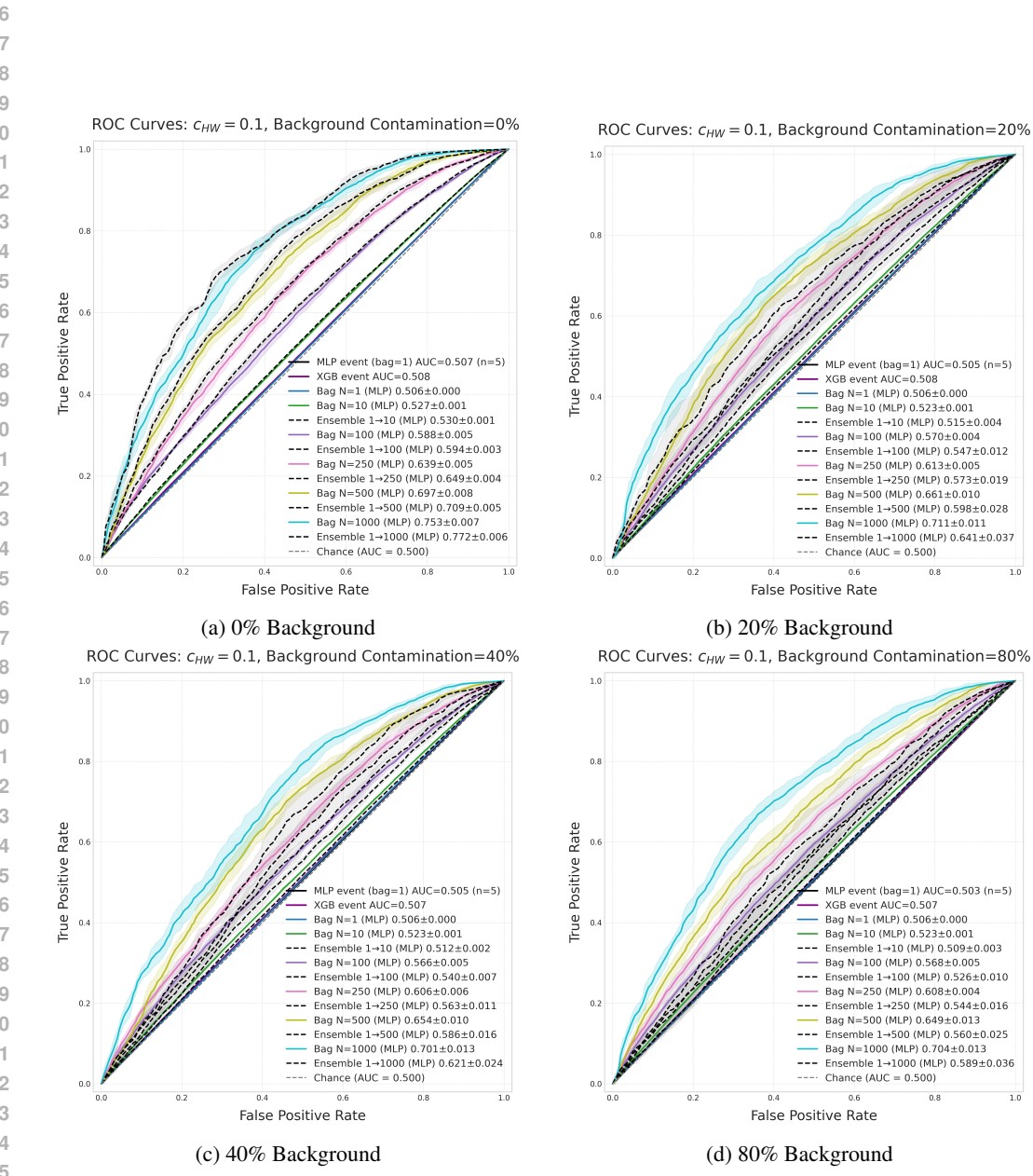

(a) 0% Background

(b) 20% Background

(c) 40% Background

(d) 80% Background

Figure 6: MIL vs. MLP: Receiver Operating Characteristic (ROC) curves for five individual binary classifiers, evaluated at various background contamination levels. The number of signal events is held constant while the total bag size increases with contamination level.

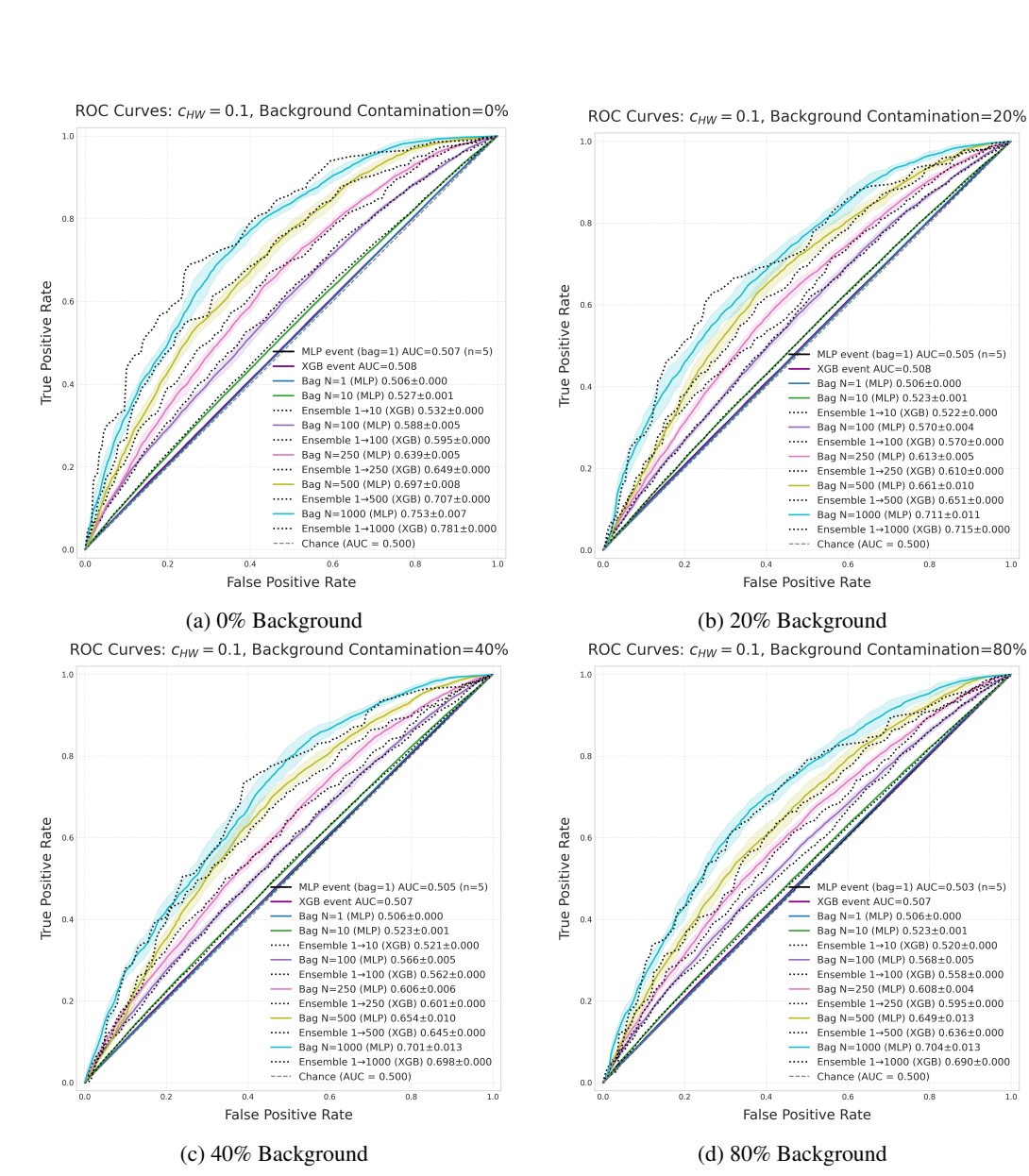

(a) 0% Background

(b) 20% Background

(c) 40% Background

(d) 80% Background

Figure 7: MIL vs. XGBoost: Receiver Operating Characteristic (ROC) curves for five individual binary classifiers, evaluated at various background contamination levels. The number of signal events is held constant while the total bag size increases with contamination level.

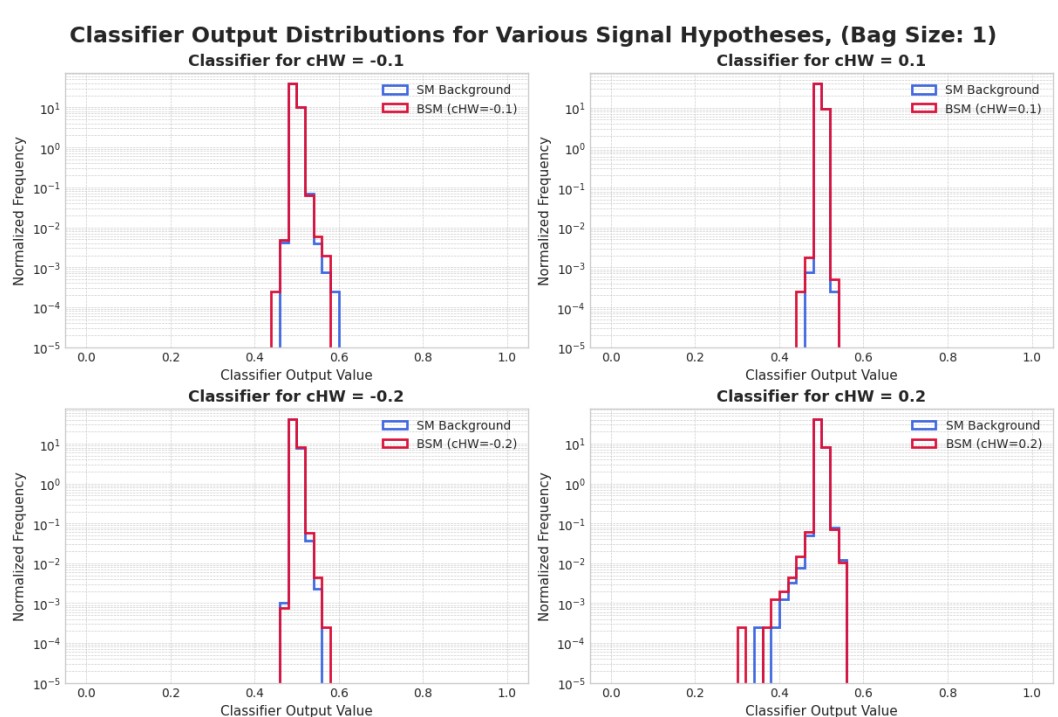

Figure 8: Distributions of the ensemble classifier output for event-by-event classification at selected $c_{HW}$ values.

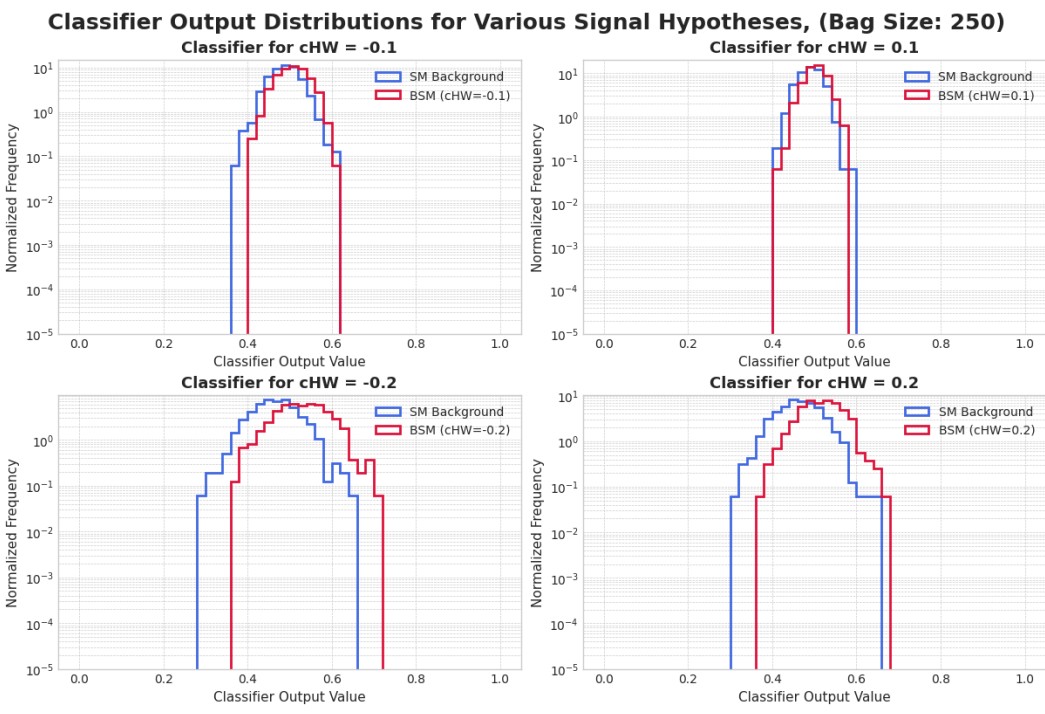

Figure 9: Distributions of the ensemble classifier output for set-based classification at selected $c_{HW}$ values.

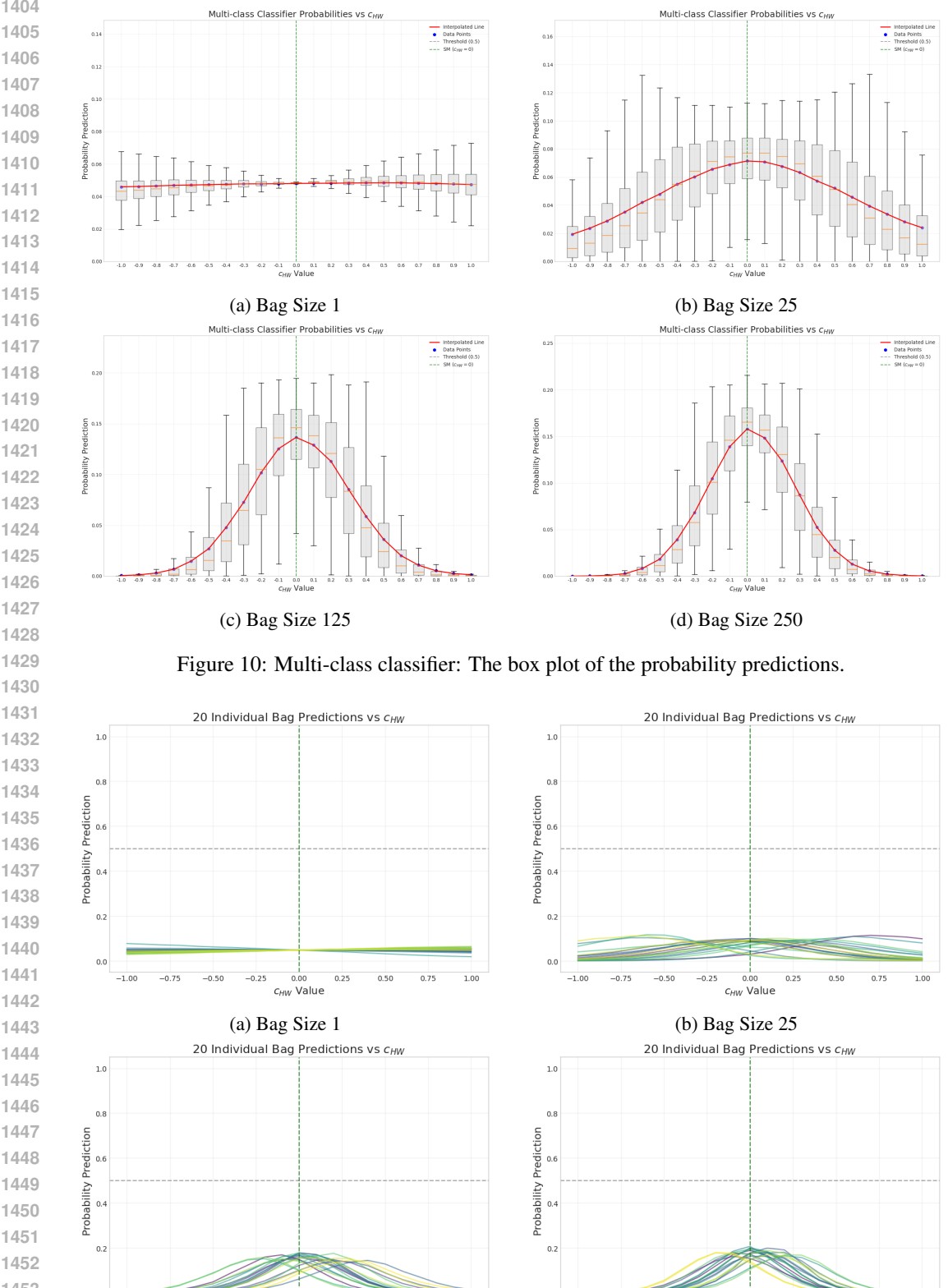

Figure 10: Multi-class classifier: The box plot of the probability predictions.

Figure 11: Multi-class classifier: Probability predictions of 20 *individual bags*.

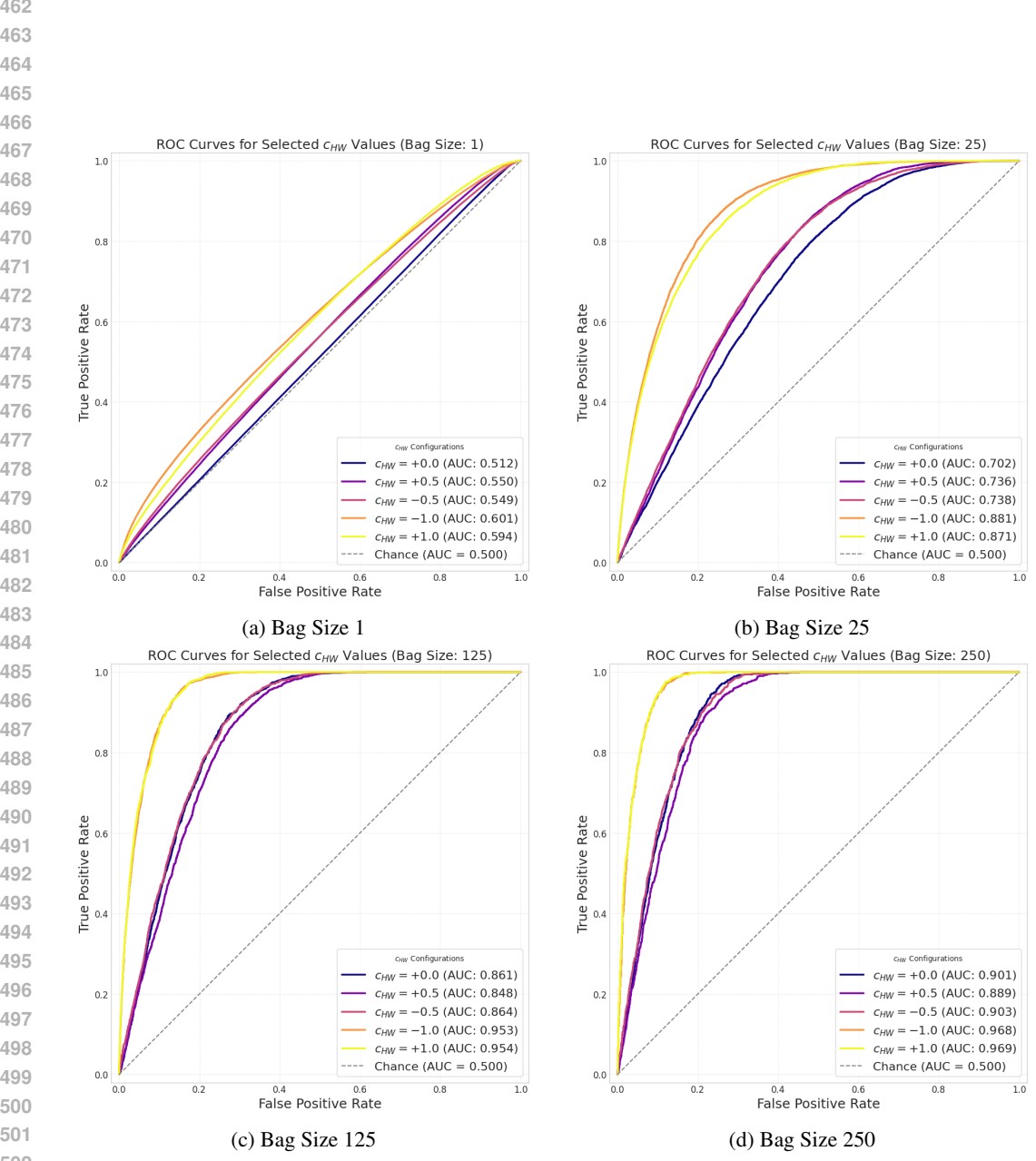

(a) Bag Size 1

(b) Bag Size 25

(c) Bag Size 125

(d) Bag Size 250

Figure 12: Multi-class classifier: ROC curves for selected $c_{HW}$ values.

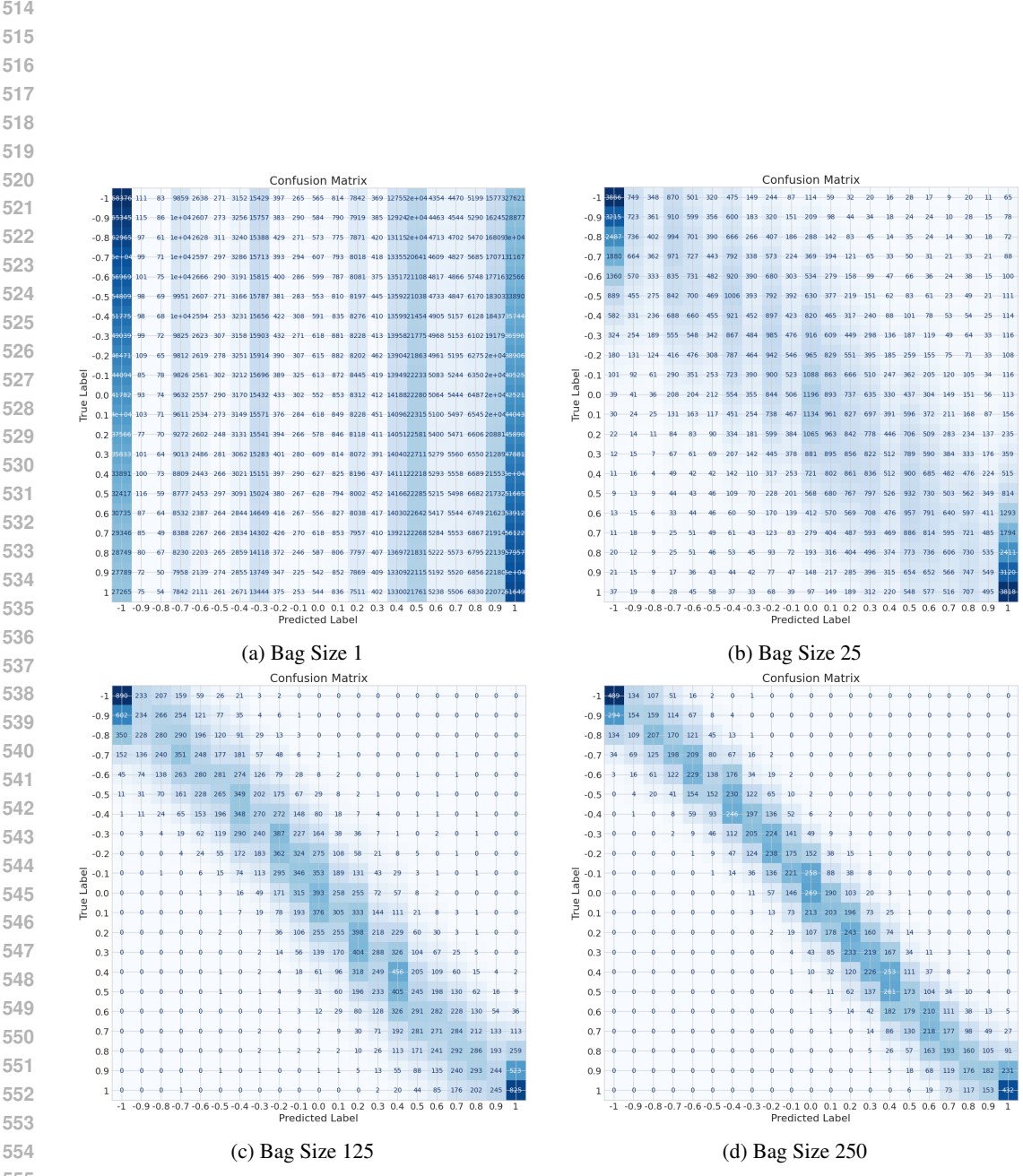

Figure 13: Multi-class classifier: The confusion matrices. As shown in Figures 14 and 15, the striped pattern in the confusion matrices did not have a profound impact on the log-likelihood ratio calculations.

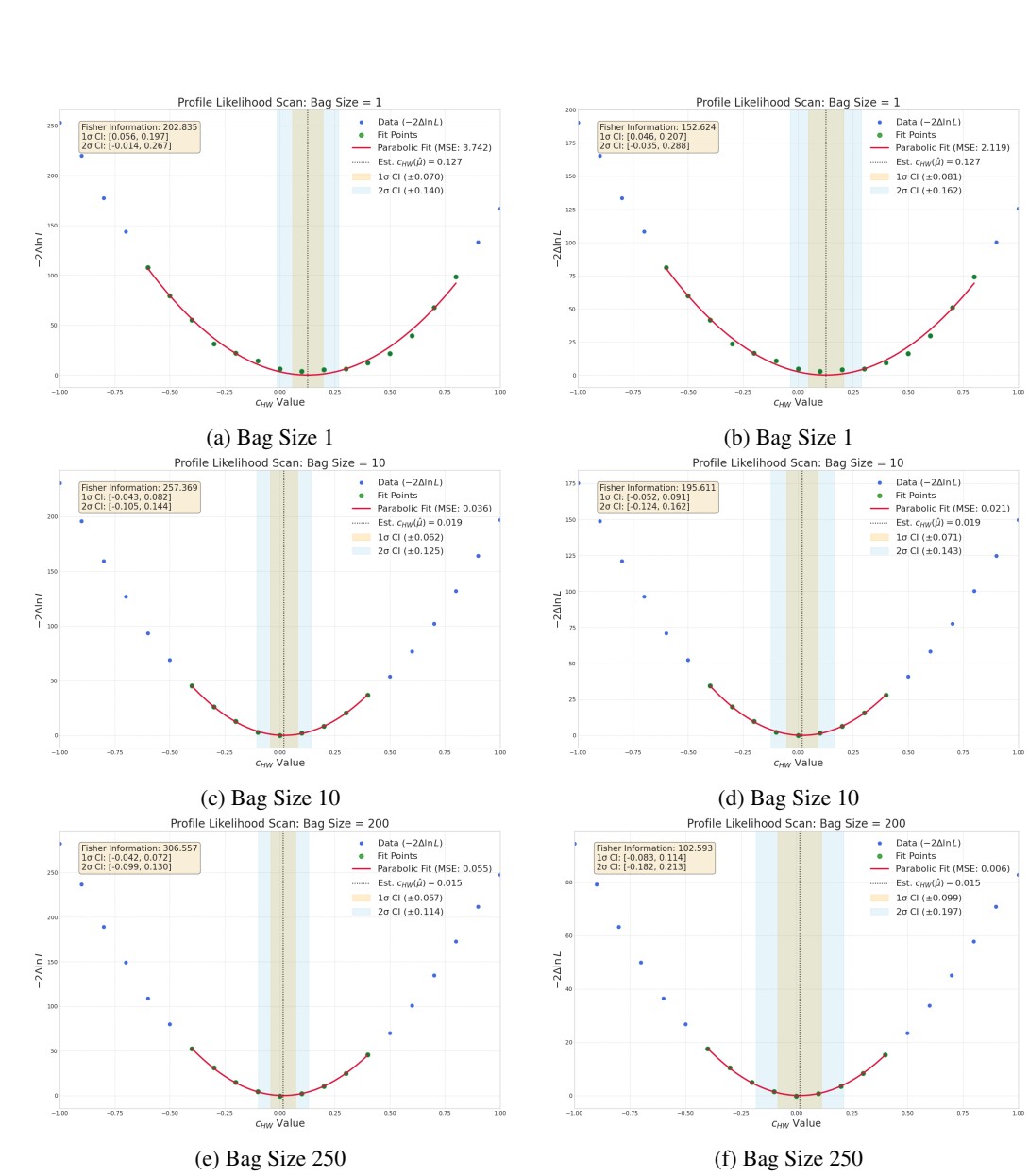

(a) Bag Size 1      (b) Bag Size 1

(c) Bag Size 10      (d) Bag Size 10

(e) Bag Size 250      (f) Bag Size 250

Figure 14: Multi-class classifier: Example of the confidence interval calculations, comparing the results before (right panels) and after (left panels) curvature calibration for the same 1000-event pseudo-experiment. As it is explained in Appendix C.4, since profile of likelihood is not perfectly smooth, number of fit points for bag size 1 is slightly larger to get a better estimate and increase its overall performance across all calculations.

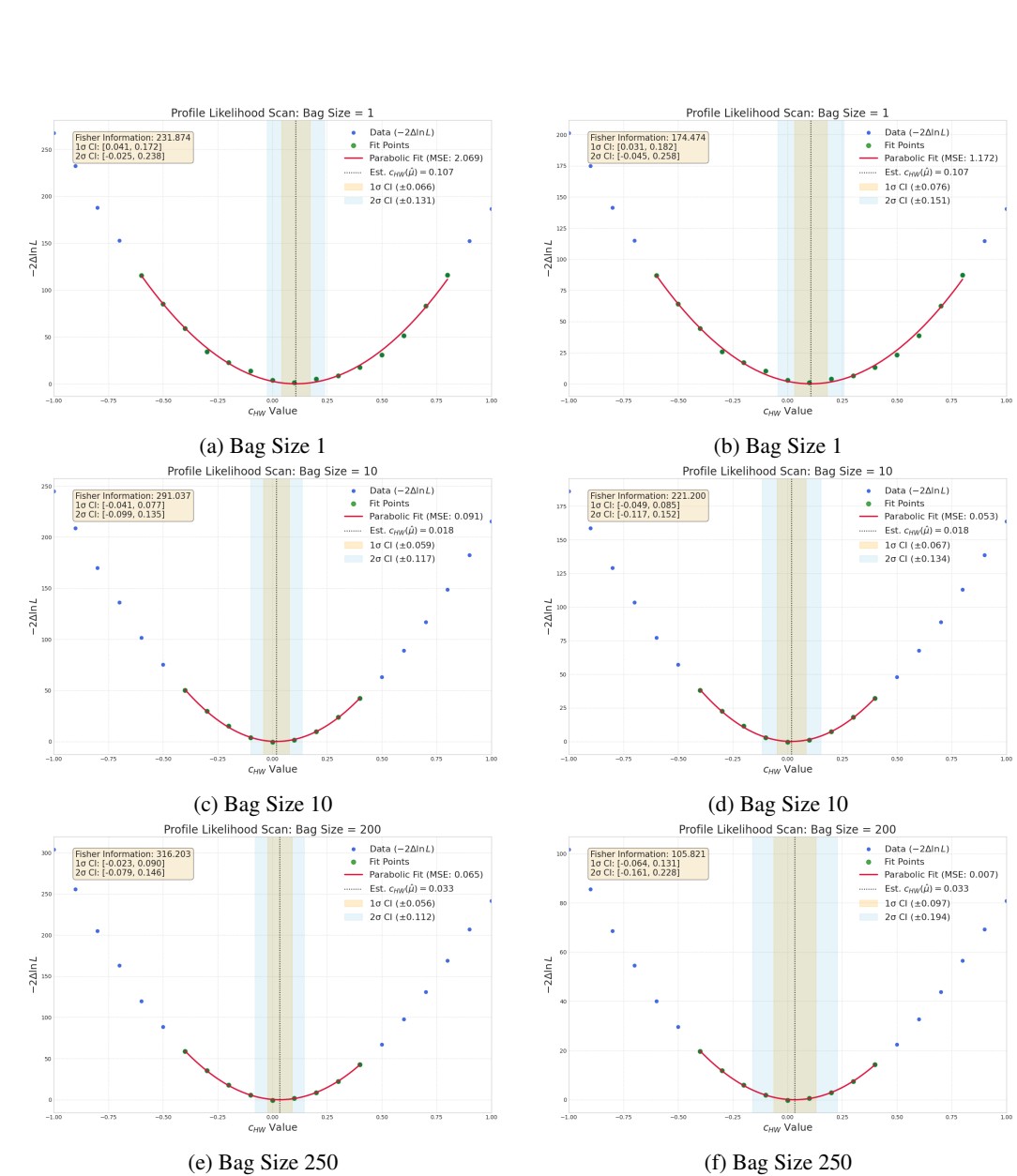

Figure 15: Multi-class classifier: Another example of the confidence interval calculations, comparing the results before (right panels) and after (left panels) curvature calibration for the same 1000-event pseudo-experiment.

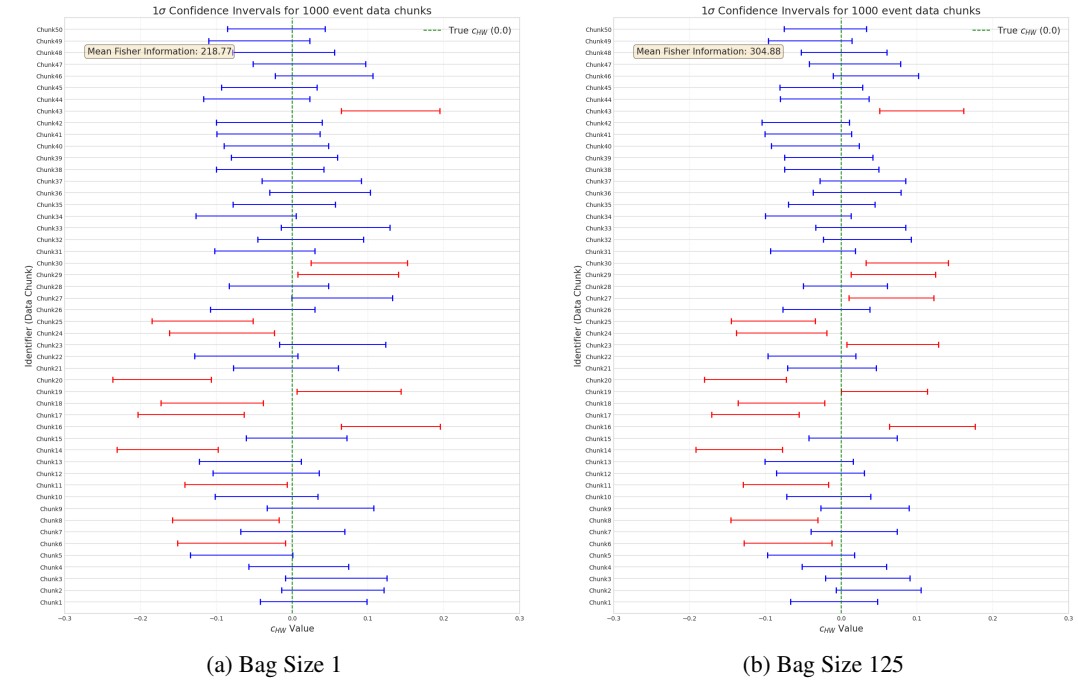

(a) Bag Size 1          (b) Bag Size 125

Figure 16: Multi-class classifier: Confidence interval coverages, with 50 of the 200 total pseudo-experiments shown. Since $\frac{\sigma_{125}}{\sigma_1} = \sqrt{\frac{\mathrm{Var}_{125}(\hat{\theta})}{\mathrm{Var}_1(\hat{\theta})}} \approx \sqrt{\frac{I_1(\hat{\theta})}{I_{125}(\hat{\theta})}} = 0.847$, this shows that increasing the bag size from 1 to 125 yields an approximately $15.3\%$ tighter constraint.

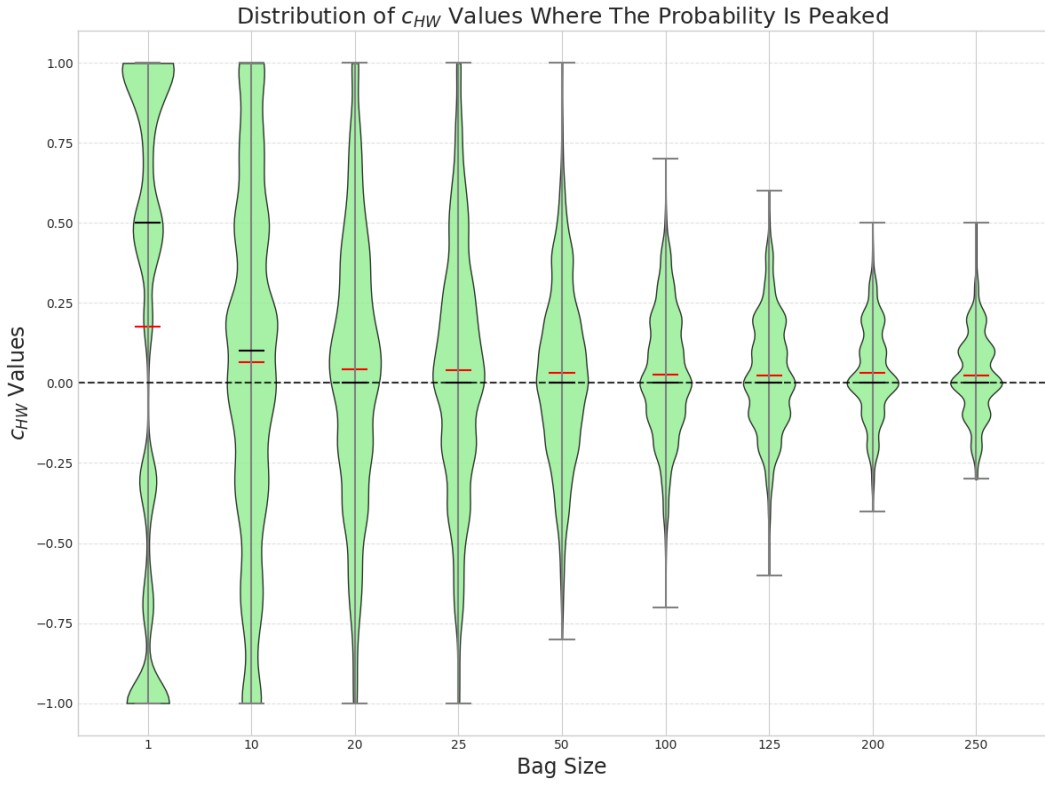

Figure 17: Violin plots showcasing the distribution of discrete $c_{HW}$ values ($\pm 0.1$ $c_{HW}$) where the predicted probability is the highest.

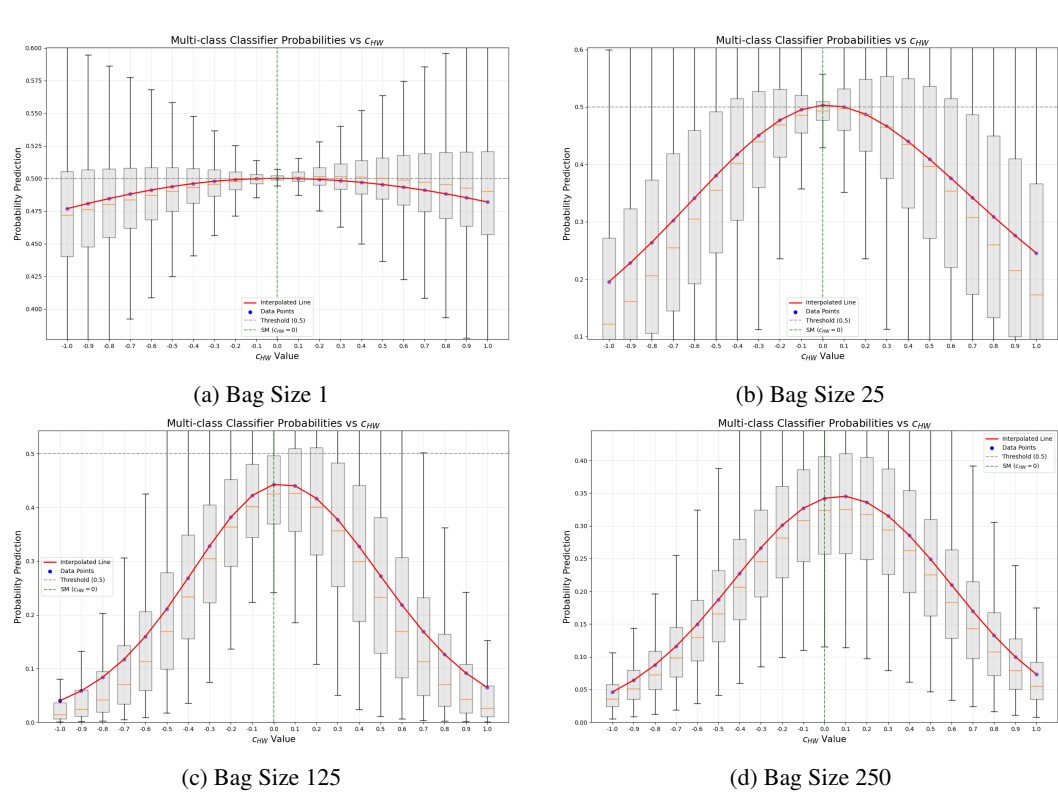

Figure 18: Parameterized Neural Network: The box plot of the probability predictions. As it's shown, the probability predictions for SM kinematics systematically decrease as the bag size increases.

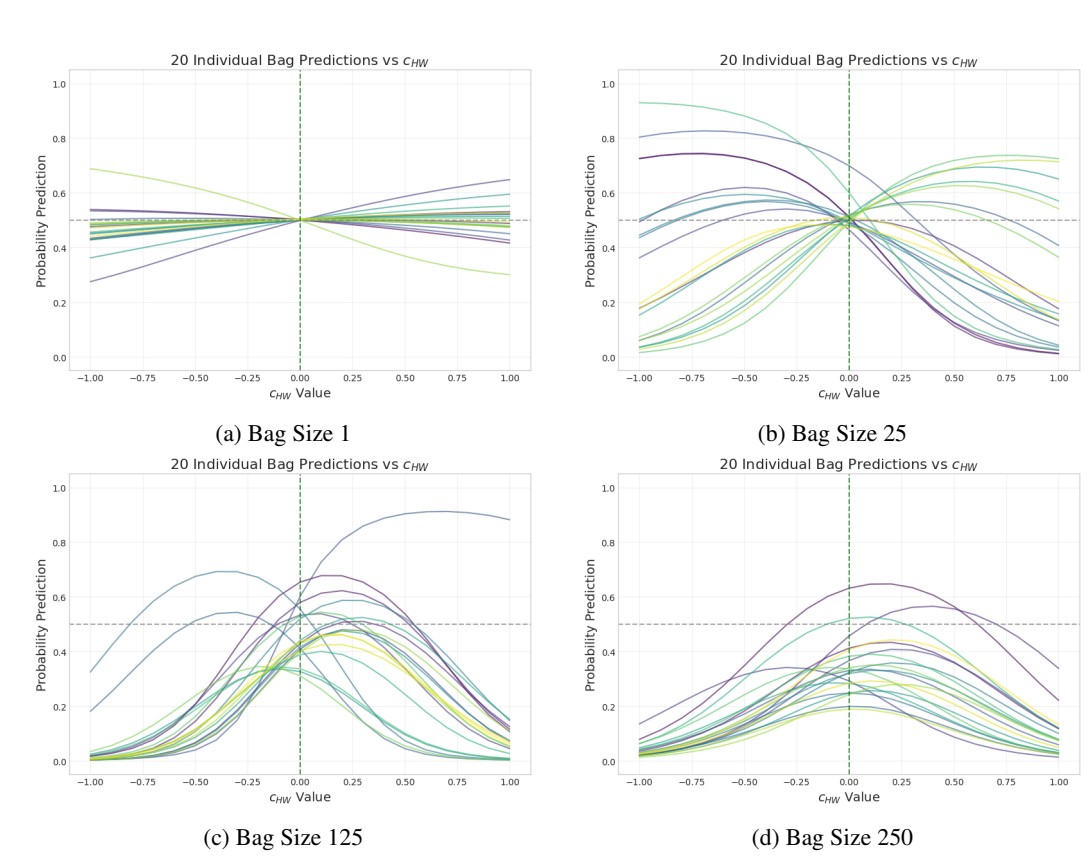

(a) Bag Size 1

(b) Bag Size 25

(c) Bag Size 125

(d) Bag Size 250

Figure 19: Parameterized Neural Network: Probability predictions of 20 *individual bags*.

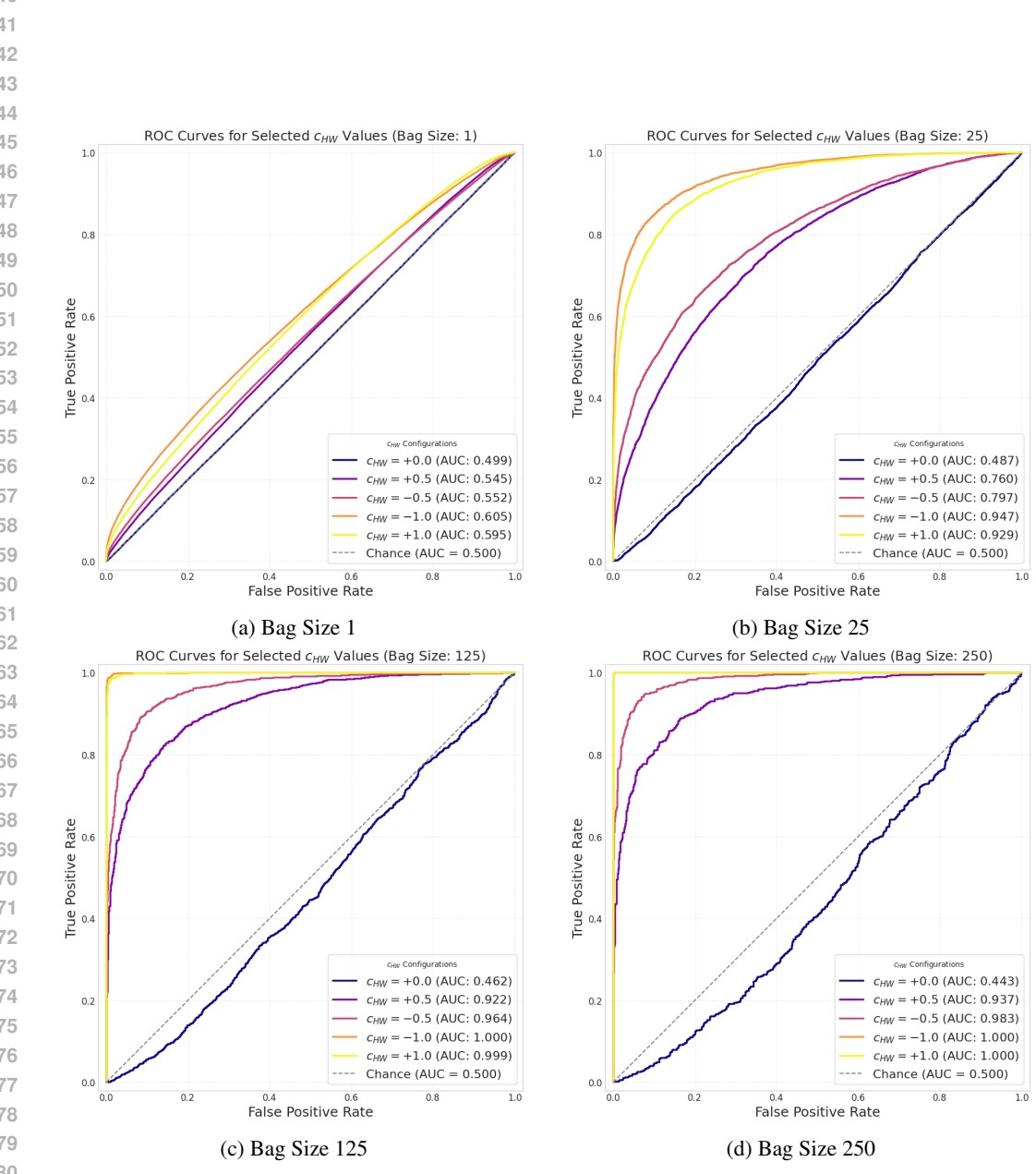

Figure 20: Parameterized Neural Networks: ROC curves for selected $c_{HW}$ values.

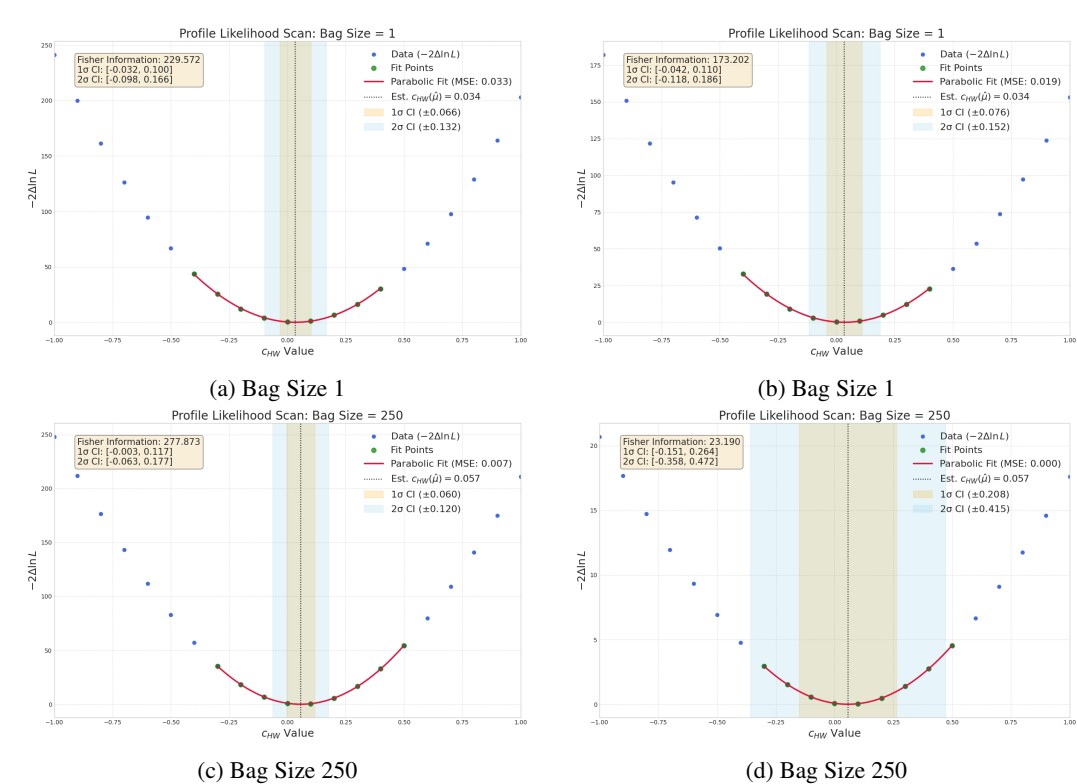

(a) Bag Size 1

(b) Bag Size 1

(c) Bag Size 250

(d) Bag Size 250

Figure 21: Parameterized Neural Network: Example of the confidence interval calculations on the same data. Right side shows before calibration, left side shows after calibration.

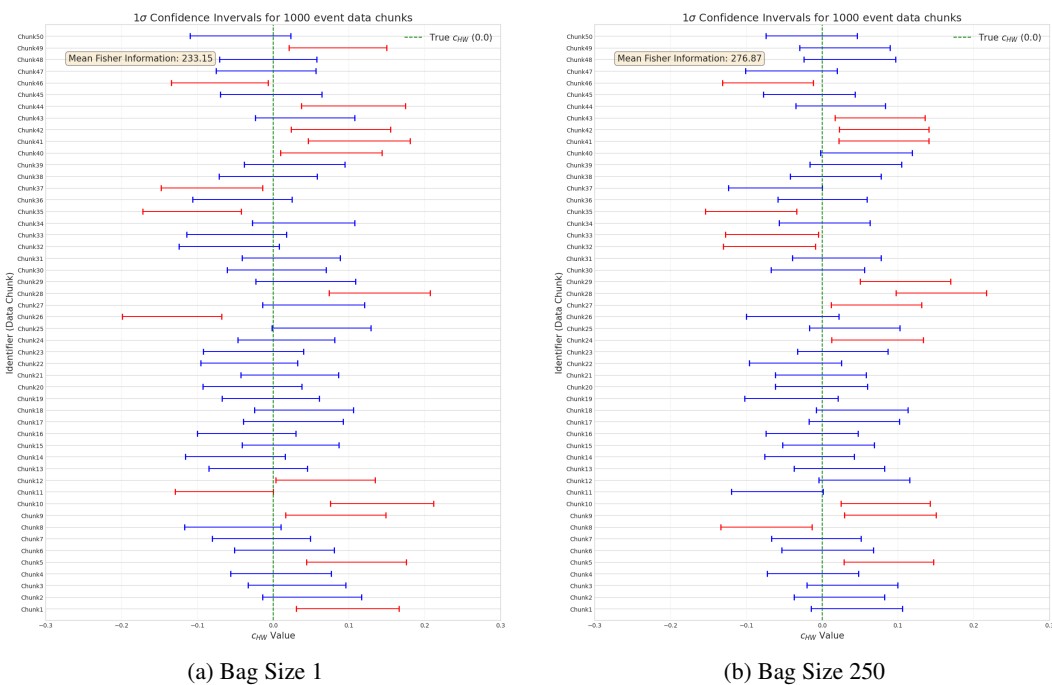

(a) Bag Size 1

(b) Bag Size 250

Figure 22: Parameterized Neural Network: Confidence interval coverages, with 50 of the 200 total pseudo-experiments shown.

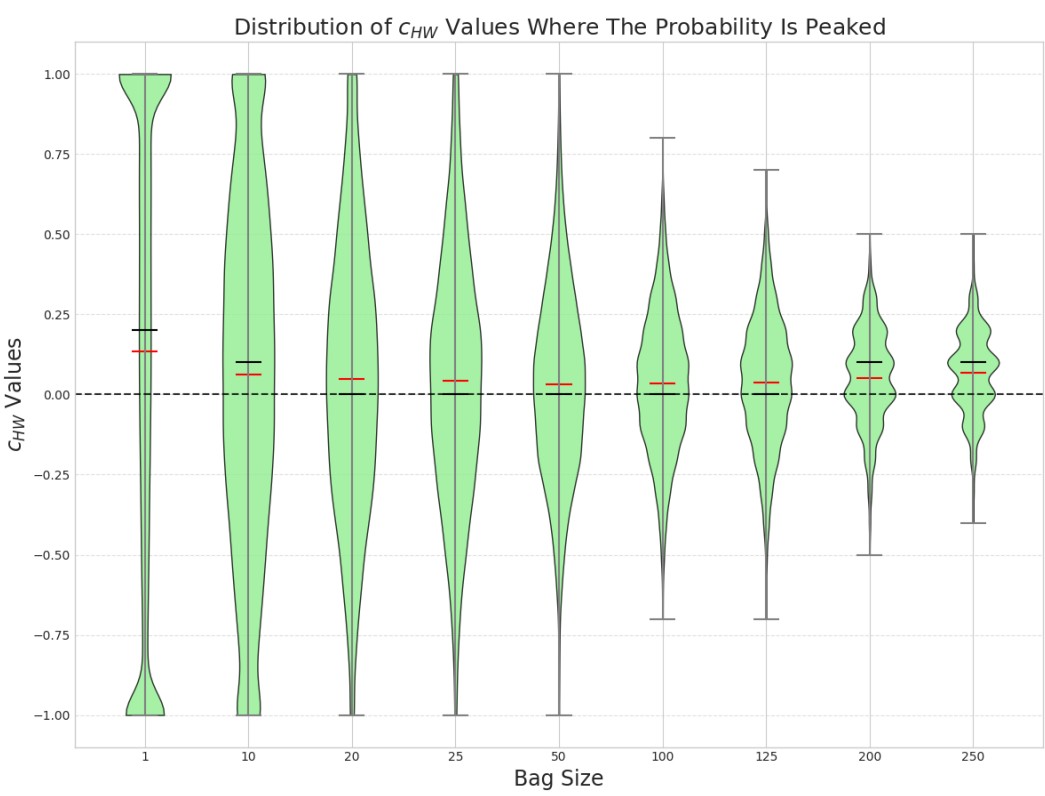

Figure 23: Parameterized Neural Network: Violin plots showcasing the distribution of discrete $c_{HW}$ values ($\pm 0.1$ $c_{HW}$) where the predicted probability is the highest.

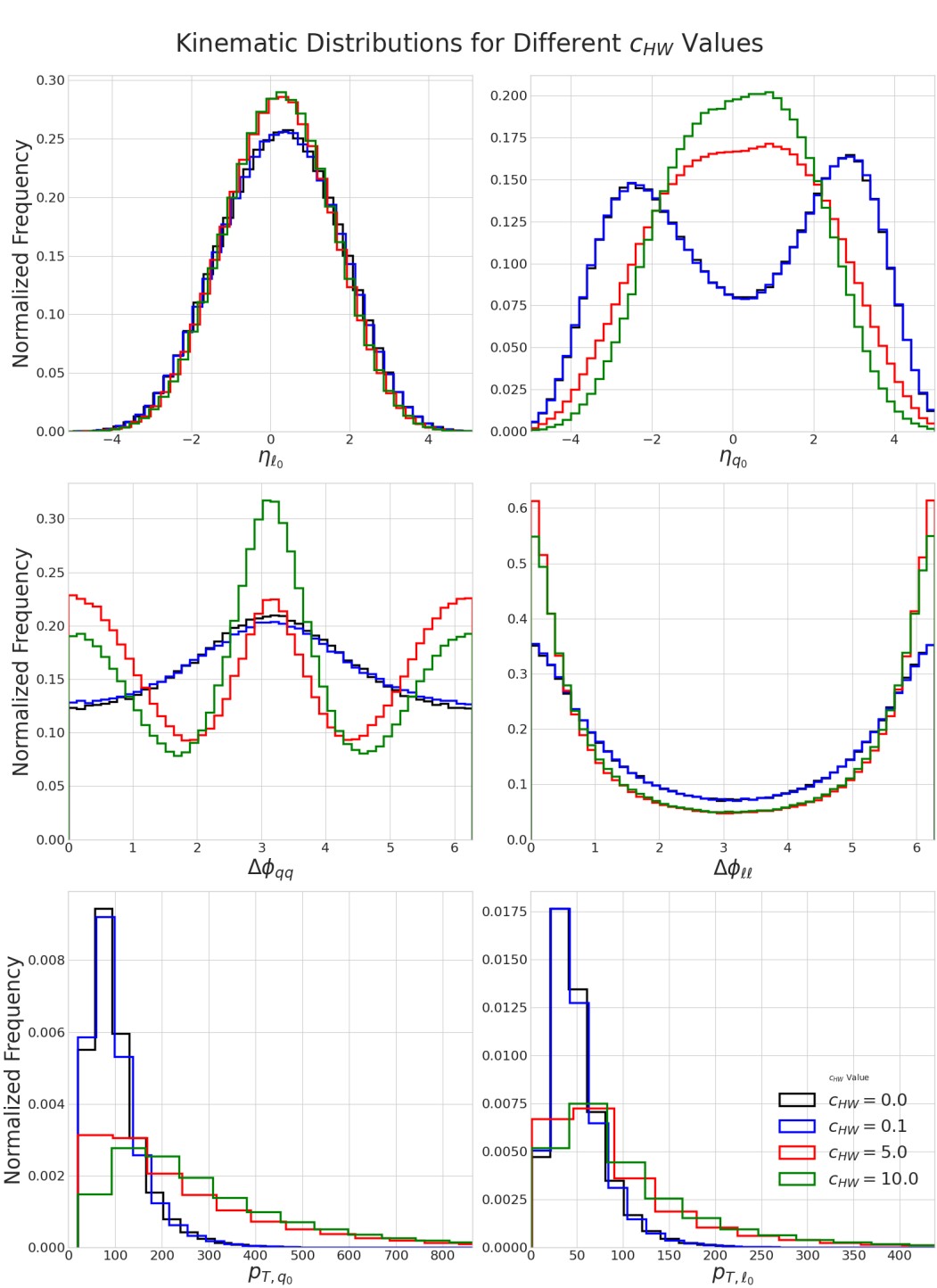

Figure 24: The dataset: A selection of kinematic distributions, comparing the Standard Model (SM, $c_{HW} = 0.0$) to various SMEFT signals. Note the nearly perfect overlap between the SM and the weak signal ($c_{HW} = 0.1$) distributions, which motivates the need for the advanced statistical aggregation method presented in this work.

