# OpenReview forum: "Increasing Information Extraction in Low-Signal Regimes via Multiple Instance Learning"
_ICLR.cc/2026/Conference — Submitted to ICLR 2026_

### Official Review · Reviewer_wFd7 · 2025-10-31

**Soundness:** 3
**Presentation:** 3
**Contribution:** 2
**Rating:** 4
**Confidence:** 3

**Summary:**

The authors propose using a Multiple Instance Learning framework to improve parameter estimation in scientific analyses in low signal regimes. They provide an information-theoretic motivation, arguing that by aggregating multiple independent instances into a 'bag', the SNR of the learning task is effectively increased. This does allow a machine learning model to extract more Fisher Information than it could from processing each instance individually. The method is demonstrated by applying it to a high-energy physics problem.

**Strengths:**

The paper is well-written and the information-theoretic motivation is sound and clearly described in the Reviewers' opinion. The empirical demonstration of performance degradation in single-instance models versus the MIL approach in the presence of background contamination is also a well-executed and easy to follow experiment.
Additionally also the discovery that models violate the second Bartlett identity and the suggested fix are interesting results by itself.

**Weaknesses:**

- The paper focuses on a single application only. While the application problem seems important, given that the authors introduce a general framework, the Reviewer would expect at least one other examples to show the applicability of the approach. Especially as the authors claim a general-purpose framework, this should also be reflected in the experiment section.
- The technical contribution of the paper seems to be limited, the core architecture is a simple MLP, and the multiple instance learning aggregation is a standard global average pooling of embeddings. There are currently no novel architectures, loss functions or training procedures proposed although the authors mention this as an important next step. From the Reviewers' perspective, the paper seems to be an exploration of a known model limitation rather than the introduction of a novel learning paradigm.

**Questions:**

See weaknesses and:
-The multi-class results were achieved by creating a large ensemble of 20 independently trained models. What would happen for a lower number of models and does this suggest a reduced robustness of a single model ?

---

> ### Author Response · Authors · 2025-11-21
>
> We thank the reviewer for their careful reading and valuable feedback. We perceive that the main reservation regarding our work stems from a view that it applies standard Multiple Instance Learning (MIL) to a specific scientific domain, i.e. High-Energy Physics (HEP), with limited methodological novelty. For clarity, our responses will be grouped by the Discussion Topics (DTs) or by the Questions (Q):
>
> > DT-1 | Clarifying the primary objective of the research:
>
> The reviewer correctly identifies lack of exploration of other ML model architectures or loss functions. However, we emphasize that our work is not merely an application paper; it is an **information-theoretic study of the breakdown of neural estimation in low-signal regimes.**
>
> This paper is not about finding a clever engineering solution to a specific problem; it is about theorizing the behaviour of neural estimators and empirically validating those predictions in controlled environments. In this context, **the simplicity of the architecture is a deliberate methodological control, not a limitation.** By fixing the architecture to a basic MLP, we isolate the performance gains attributable strictly to the set-based aggregation strategy and the resulting Fisher Information (FI) scaling.
>
>
> > DT-2 | Empirical evidence on a single field:
>
> We selected this problem specifically because it represents a generic Hypothesis Testing problem with prohibitively low Signal-to-Noise Ratio (SNR). Unlike many real-world datasets, the high fidelity physics simulation allows us to precisely control the signal strength. This makes it an ideal environment to empirically validate our theoretical derivations (Eq. 15). While we agree that future work should explore other domains (e.g., genomics), *the current setup is sufficient to validate the general theoretical claims* regarding SNR scaling and information extraction.
>
> > DT-3 | On technical contributions:
>
> Below, we clarify the core contributions that we believe were overlooked, specifically in the context of our field, and the development of the effective FI:
>
> It is crucial to contrast our results with the prevailing wisdom in our field, which argues that single-instance classifiers are sufficient for optimal inference on i.i.d. data because the likelihood of a set factorizes. [1] We note that providing counter-examples to prevalent notions in a research field is a critical contribution to the literature.
>
> We hypothesized that single-instance learners would perform sub-optimally in low-signal regimes, and theorized that MIL can be utilized as a simple artificial SNR increasing mechanism and resist performance degradation as SNR decreases. **We performed controlled experiment and validate this SNR perspective.** We further study sub-optimality in the context of Fisher Information and developed a mathematical framework for characterizing the information extraction, as shown in Eq. 15:
>
> $I_{\text{eff}}(\theta) =  \frac{I_{true}(\theta)}{1 + \frac{ \sigma^2_{\epsilon}(N_B)}{ N_B I_1(\theta_0) (\Delta\theta)^2}}$,
>
> To our knowledge, the $I_{\text{eff}}(\theta)$ we derived is a **novel mathematical development** connecting ML error variance to information recovery. Furthermore, as the reviewer noted, our identification of the second Bartlett identity violation in neural networks, and the proposed post-hoc calibration addresses a fundamental reliability issue in "AI for Science."
>
> > Q-1 | Robustness of Single Models:
>
> Yes, exactly as the reviewer predicts. As shown in Figure 5 (App. C.5), single-instance models (N=1) in this low-SNR regime produce highly non-smooth likelihood ratios. Aggregating instances into a bag (N>1) increases the SNR, resulting in smoother, more robust convergence for individual models. We used ensembles specifically to give the single-instance approach the best possible chance of competing, thereby strengthening the rigour of our comparison.
>
> We hope that this covers all of reviewer’s concerns. We are ready to answer any further questions or comments by the reviewer. Since the paper's primary value lies in exploring the statistical limits of neural estimators in low-signal conditions, rather than simply applying MIL to a physics dataset; we kindly ask the reviewer to reconsider the score in light of these theoretical and methodological contributions.
>
>
> [1] Benjamin Nachman and Jesse Thaler. Learning from many collider events at once. Phys. Rev. D, 103:116013, Jun 2021. doi: 10.1103/PhysRevD.103.116013. URL https://link.aps.org/doi/10.1103/PhysRevD.103.116013.

---

> ### Author Response · Authors · 2025-11-28
>
> As the discussion period is approaching its end, we wanted to respectfully check if our previous response and the revised manuscript have addressed all of your concerns. Specifically, we hope our response clarified the main critisms such as the choice of application, and the technical contribution & the objective of the paper.
>
> We remain available to provide any further details you might need. We would greatly appreciate your feedback and hope these clarifications might warrant a reconsideration of the score.

---

### Official Review · Reviewer_PHoD · 2025-10-31

**Soundness:** 3
**Presentation:** 3
**Contribution:** 2
**Rating:** 2
**Confidence:** 3

**Summary:**

This paper discusses a hypothesis testing scenario, where evaluating the likelihood function is intractable and hence the standard likelihood ratio test (LRT) becomes infeasible. The paper considers an alternative approach, where a neural classifier is trained based on simulated data under different hypotheses. In particular, it considers a solution where multiple instances are assigned to one common label (called MIL). The paper examines this idea in particle physics, for detecting deviations from Standard Model in collision experiments. By experimentation on synthetic data, it is shown that the proposed method is superior to the combination of decisions on individual instances (ensemble methods). This observation is further theoretically justified by arguments involving Fisher information and Cramer-Rao bound.

**Strengths:**

I am not familiar with physics literature and hence cannot assess the significance of the paper within this field. From a general statistics perspective, especially in the context of data fusion, the contribution of the paper is a lightweight method, based on LRT, which fuses multiple instances at a feature level rather than at a decision level. Feature-level fusion is known to be superior to decision-level fusion, especially in low-SNR regimes, but is generally considered a complex task.

**Weaknesses:**

I wonder how novel or substantial contribution is. As already mentioned, the fact that feature-level fusion is superior to the decision level is well-known and intuitive. The use of NNs for estimating likelihood ratios is not entirely new and is extensively discussed in the context of neural ratio estimation (NRE). From the perspective of MIL, the paper considers a simplified scenario, which to me is a repetition of the
standard point estimation theory with multiple observations. A major part of the theoretical discussions, e.g. the vanishing of ML error and growth of FI with O(\sqrt{N}), can be found in multiple classical sources.

Another drawback of the suggested approach is that for deployment, it requires an ensemble of independent observations of similar size to the ones used for training. This can be a limitation in practice.

The presentation of the paper can also be improved. It is sometimes difficult to understand the motivation behind the concepts introduced. For example, I am not familiar with the notion of effective Fisher information, and it is not clear to me what it implies. Some notations remain unexplained too. For example, in line 198 e_ij seems to refer to the elements of e_i, but this is not defined. Moreover, \theta_SM and \theta_SMEFT are not properly introduced.

**Questions:**

After understanding the problem of interest, I am surprised about the use of LRT, in this context. The reason is that one of the alternatives is presented as a composite hypothesis (theta\neq 0). A consequence of applying LRT is that the alternative values of theta (e.g. \theta_1) must be selected beforehand. How can this be done? And are not tests such as GLRT more suitable for this scenario?

If I understand it correctly, the training procedure does not explicitly bias the logits toward the individual likelihood ratios. Is it possible to guarantee that they are estimates of LRs? Indeed, Appendix C shows that they are biased in nature. And if biased, how is the presented theory based on CRB relevant to them?

As a minor comment in (11), do you mean by “+ 0” a higher order term?

---

> ### Author Response · Authors · 2025-11-21
>
> We thank the reviewer for their detailed comments and insightful connections to feature-level fusion and NRE. We believe the primary reason for the unclear novelty stems from the different languages used in these literatures. With this response, we hope to clarify the questions and comments raised by the reviewer. For clarity, our responses will be grouped by the Discussion Topics (DTs):
>
> > DT-1 | Problems regarding presentation, i.e. $\theta_{SM}$ & $\theta_{SMEFT}$, the  “+ 0” term in Eq. 11, and confusion in line 198 "$e_{ij}$":
>
> The $\theta_{SM}$ & $\theta_{SMEFT}$ are defined in line 151, and the “+ 0” term in Eq. 11 comes from the expectation value stated in line 212. Each feature $\bar e_{i}$ of the vector $\mathbf{\bar e}$ is created by taking the mean of the feature values across the number of events dimension. We acknowledge this notation is misleading, and we thank the reviewer for catching this. We will make the necessary adjustments for overall clarity.
>
> > DT-2 | On the deployment drawback:
>
> We developed "Dynamic-bagging" data augmentation method for the training. At each epoch, the events within the training set are randomly shuffled and re-grouped into new, unique bags. This training procedure is discussed further in Appendix B.2.2, and is not required for deployment.
>
> > DT-3 | On classical results:
>
> Under classical asymptotic theory, the ***SNR*** scales with $\sqrt{N}$ (the FI stays constant for fixed dataset) and estimation error is negligible. However, our analysis focuses on the *pre-asymptotic regime* where ML models are suboptimal. In this regime, the ML-induced error term is non-negligible. We clarify this distinction further in DT-5.
>
> > DT-4 | On the Generalized Likelihood Ratio Test (GLRT):
>
> Our analysis uses both LRT and GLRT. Since binary classification is a simple-vs-simple test (i.e. $H_0: \theta = 0$ vs $H_1: \theta = 0.1$) LRT is the optimal test statistic. Our multi-class and parameterized networks are NN-based methods for constructing the profile likelihood, which is the central component of the GLRT. Profiling the log-likelihood ratio curves to find the minimum is the standard CERN analysis procedure and is statistically equivalent to the GLRT.

---

> > ### Author Response · Authors · 2025-11-21
> >
> > > DT-5 | Novelty, Fisher Information, CRB, Data Fusion:
> >
> > Utilizing the data collected from multiple sources through different means is a fundamental idea and has various names in different fields: multiple-instance learners, feature/decision fusion (which we now also referenced in the paper), and per-ensemble models etc. Nevertheless, prominent figures from Berkeley & MIT argue against its use in their influential paper, [1] suggesting that single-instance methods are sufficient and that MIL introduces unnecessary training complexity. This paper provides a theoretical and empirical counter-example to this notion, which is a a critical contribution to the literature.
> >
> > The small bias, the violation of the second Bartlett identity, and the effect of post-hoc procedures addressing this issues on our analysis have been thoroughly discussed in App. C.2.1 & C.2.1. Our theory is not based on the assumption that NNs achieve CRB, but on the fact that they ***fail*** to. As detailed in Sec. 3.2 and App. A, we show that *even if the model is unbiased*, there would be some term related to ML error which can prevent proper extraction of the FI latent in the data. The information contained in the data (true FI) and the information that can be obtained through ML models (effective FI) are two separate notions.
> >
> > **Our contributions:** We predicted that ML models would fail reach optimal classifier level in low-signal regime. We then utilized MIL as a simple artificial SNR increasing mechanism in High-Energy Physics (HEP), and demonstrated that MIL does indeed resist performance degradation as we go to the lower signal regions. (**Important for HEP**) We then developed a mathematical framework for characterizing the information extraction:
> >
> > $I_{\text{eff}}(\theta) =  \frac{I_{true}(\theta)}{1 + \frac{ \sigma^2_{\epsilon}(N_B)}{ N_B I_1(\theta_0) (\Delta\theta)^2}}$,
> >
> >
> > and demonstrated that the increase in performance can be predictable under a certain set of conditions. To our knowledge, $I_{\text{eff}}(\theta)$ we derived is a **novel mathematical development.** Furthermore, in our analysis, we made an observation of, and post-hoc solution to the violation of second Bartlett identity, which, to our knowledge, is also **novel**.
> >
> > **Low vs high signal regions:** In high signal region the error term is negligible, therefore $I_{\text{eff}}(\theta) \approx I_{true}(\theta)$; but in the low signal region, the information contained in a single sample is low and the variation of the ML error is high, therefore the ratio $\frac{ \sigma^2_{\epsilon}(N_B)}{ N_B I_1(\theta_0) (\Delta\theta)^2}$ becomes non-negligible. Since MIL artificially increases the SNR, the increase in signal available per decision helps ML model to better discern the small differences and lower the resulting error term. Furthermore, under specific conditions, which are thoroughly discussed in Sec. 3.2, App. A, and C.6, and empirically observed in Sec. 4.2, *one may be able to predict this increase in performance as they increase the input size.*
> >
> > We hope this covers all of reviewers concerns. If further clarification needed, we are ready to answer them. In light of this, we kindly ask reviewer to reevaluate the novelty criticism, and hope that the reviewer would consider increasing their score.
> >
> > [1] Benjamin Nachman and Jesse Thaler. Learning from many collider events at once. Phys. Rev. D, 103:116013, Jun 2021. doi: 10.1103/PhysRevD.103.116013. URL https://link.aps.org/doi/10.1103/PhysRevD.103.116013.

---

> ### Author Response · Authors · 2025-11-28
>
> With the discussion period ending soon, we are writing to follow up on our rebuttal to ensure we have adequately addressed all your concerns and your reservations regarding the technical novelty of the work.
>
> In our response, we strove to highlight that the paper's core value lies not just in the application, but in the information-theoretic derivation of SNR scaling (Eq. 15), the identification of the Bartlett identity violation in neural estimators, and providing theoretical and empirical counter-examples to prevalent notions in a research field.
>
> We would value your feedback on whether our response, along with the revisions in the manuscript, help resolve your concerns. We hope these clarifications might warrant a reconsideration of the score. We are happy to engage further if any points remain unclear.

---

### Official Review · Reviewer_FvWc · 2025-11-01

**Soundness:** 3
**Presentation:** 3
**Contribution:** 3
**Rating:** 8
**Confidence:** 4

**Summary:**

This paper demonstrates and theoretically justifies that multiple instance learning (MIL) for parameter estimation with i.i.d. data can outperform single instance learning (SIL) at low signal-to-noise ratios. Prior work demonstrated that single-instance learning is sufficient, assuming the training is optimal, but for low signal-to-noise, this may not be satisfied. They demonstrate the effectiveness of MIL in constraining Wilson coefficients of the standard model effective field theory (SMEFT) using kinematic information from subatomic particle collision events at the CERN LHC, observing that, for low signal-to-noise ratios, pooling instances can increase the effective Fisher information compared to single-instance approaches.

**Strengths:**

* Investigation of multiple instance learning in a new setting (low signal-to-noise ratios), which is of practical importance at the CERN LHC.
* Theoretical justification using effective Fisher information to explain why MIL practically improves on SIL for low SNRs.
* Comparison to multiple baselines, including parameterized neural networks, and in multiple settings, including binary and multi-class classification.
* Code is made public.

**Weaknesses:**

* Unclear if the data has been or will be made public.
* Some details on the training procedures are missing, e.g. how large is the training data set? How many epochs was each algorithm trained for? Were training hyperparameters, e.g., learning rate, optimized? Etc. Since part of the claims depend on (non)optimality of the models, these are important considerations.

**Questions:**

* Was min/max pooling studied in addition to the average pooling of rht embedding vectors in a given bag? Min/max pooling seems like a more suitable choice if the goal is to classify if there is *any* signal instance in the bag.
* Can the studied datasets be made public?
* Could you define the effective Fisher information or clarify how/why it differs?
* Fig. 1: How large is the training data set? Are all of these models trained with the same size data set? How many epochs was each algorithm trained for? Were training hyperparameters, e.g., learning rate, optimized?
* Would the single-instance learning eventually “catch up” to the multiple instance learning if provided a large enough dataset even in the low SNR regime? If so, then another way to cast these results are that MIL is more data efficient.
* Fig. 2: Fix typo “Ensamble”

---

> ### Author Response · Authors · 2025-11-21
>
> We thank the reviewer for their careful reading, positive assessment, and endorsement of our work. We hope to clarify and answer to the questions and comments raised by the reviewer. For clarity, our responses will be grouped by the Discussion Topics (DTs) or by the Questions (Q):
>
> > DT-1 | Availability of data:
>
> The data was created using open-source simulation software (MadGraph5_aMC@NLO v3.6.2 interfaced with SMEFTsim v3.0). The data creation procedure is explained in App. B.1, and the implementation should have been uploaded in the GitHub repository. We thank reviewer for catching this mistake, and sincerely apologize for our oversight. Detailed instructions have been uploaded to the repository, along with the bash scripts to create arbitrary amount of data with a few lines of command.
>
> > DT-2 | Details on the training procedure:
>
> Due to space constraints, training details were moved to Appendix B.2, but we realize this was not properly referenced. We again thank reviewer for the helpful comment.
>
> *Here, we summarize the relevant information:*
>
> For each physical parameter value, $10^6$ collision events were generated. 20% was held out as a final test set, and the remainder was split into training (80%) and validation (20%) sets. Since the primary objective of the paper is an information-theoretic study of the breakdown of neural estimation in low-signal regimes, the hyperparameters like number of layers/neurons etc. were not optimized as we wanted to understand the effect of the MIL in a controlled environment. But since the research was on optimality, the epoch number was practically set to infinity to accommodate arbitrary number of gradient updates. Likewise the learning rate was dynamic, and it decreases if there was no improvements observed for PATIENCE number of epochs. Before the final analysis, we gave a couple of training runs to make sure that no model was stopped before reaching its performance plateau with the parameters we set. To ensure a fair comparison between MIL vs the strong single-instance baseline model XGBoost, we used a sophisticated framework Optuna for hyperparameter optimization of the XGBoost model.
>
> > Q-1 | The min/max aggregation:
>
> Unlike classical MIL where the goal is to find a 'needle in a haystack' (where Max Pooling excels), our goal is to estimate a parameter that affects the probability density of all events. Since the log-likelihood of a set of i.i.d. events is the *sum* of individual log-likelihoods: $\ln \mathcal{L}_{bag} = \sum \ln p(x_i | \theta)$, the average pooling would allow the network to approximate this summation, whereas max pooling would discard the vast majority of the statistical power.
>
> > Q-2 | Dataset:
>
> Explained in DT-1.
>
> > Q-3 | Effective Fisher Information (FI):
>
> Since the information contained in the data (true FI) and the information that can be obtained through ML models (effective FI) may not be the same, we mathematically analyzed its relation to ML error and derived the Eq. 15:
>
> $I_{\text{eff}}(\theta) =  \frac{I_{true}(\theta)}{1 + \frac{ \sigma^2_{\epsilon}(N_B)}{ N_B I_1(\theta_0) (\Delta\theta)^2}}$,
>
> In high signal region the error term is negligible, therefore $I_{\text{eff}}(\theta) \approx I_{true}(\theta)$; but in the low signal region, the information contained in a single sample is low and the variation of the ML error is high, hence the ratio $\frac{ \sigma^2_{\epsilon}(N_B)}{ N_B I_1(\theta_0) (\Delta\theta)^2}$ becomes non-negligible. Therefore, this effectively reduces the information we extract from the data.
>
> Since another reviewer also asked the same question, we realize that this notion was not clearly conveyed our paper. We will make the necessary clarifications in the main text as well.
>
> > Q-4 | Fig. 1:
>
> Explained in DT-2.
>
> > Q-5 | Large-data limit:
>
> Under high SNR, or infinite data limit with perfect optimization, the single-instance learners (SIL) and MIL are asymptotically equivalent. However, our results indicate a practical *SNR limit* in which even with large datasets, the optimizer may fail to find a descent direction for optimal SIL. However the exact relationship between dataset size and SNR limit is unknown. We thank reviewer for this very good, and thought provoking question!
>
> > Q-6 | Typo:
>
> It will be fixed. Thank you!
>
> We thank again for the reviewer for the helpful comments, and fruitful discussions. We are ready respond to any further questions or comments raised by the reviewer.

---

> > ### Comment · Reviewer_FvWc · 2025-11-26
> >
> > Thank you, these clarifications/additions address my comments.

---

### Author Response · Authors · 2025-11-21

We thank the reviewers again for their constructive comments and feedback, which we have used to refine our manuscript. We would like to summarize the key modifications made in the revised version:

*   **Code Availability:** Added detailed data generation scripts and instructions to the anonymous GitHub repository.
*   **Contextualization:** Referenced data fusion literature in the introduction to better frame the contribution.
*   **Training Details:** Explicitly referenced the machine learning training and optimization details (Appendix B.2) within the Results section.
*   **Theoretical Clarification:** Expanded the Theory section (Section 3.2) to clearly distinguish between *effective* Fisher Information ($I_{eff}$) and *true* Fisher Information ($I_{true}$), and elaborated on the asymptotic behaviour of neural estimators in low- vs. high-signal regimes.
*   **Corrections:** Fixed notation inconsistencies and minor typos.

We remain available to respond to any further questions or comments during the discussion period.

---

### Meta-Review · Area_Chair_iFcX · 2026-01-08

**Summary:**

This paper investigates Multiple Instance Learning (MIL) for parameter estimation in low–signal-to-noise regimes. The authors motivate the method with an application in high-energy physics, and provide both empirical evidence and an information‑theoretic argument on why aggregating 'bags' can improve the estimation in terms of  Fisher information relative to single‑instance learning (SIL). After a thorough rebuttal period, there is agreement that experiments are competent (with some details clarified) and that the framing via an info theory lens is interesting, but core concerns regarding novelty and cope remain substantial enough to warrant further round of review.

**Reviewer Concerns:**

Two of the reviews note that a central idea that feature‑level fusion can outperform decision-level in these low-signal settings is established in classical literature as well as more recent areas like data fusion. From this view, it is possible that much of the paper's theoretical component is an interesting but limited re-articulation of such general results, focused on a single application domain. Some of the thorough author rebuttal ameliorates concerns but does not fully resolve issues related to novelty.

**Reviewer Scores:**

The reviewer scores span a range and are mostly consistent with the content of the reviews. However the positive review lacks detail and does not engage with the issues identified by the other reviewers, instead listing things such as 'comparison to multiple baselines' and 'code is made public' in the strengths which do not indicate scrutiny of the specific paper and are rather generic checklist items that any paper should contain

---

### Decision · Program_Chairs · 2026-01-26

Reject